# Respiratory supercomplexes act as a platform for complex III-mediated maturation of human mitochondrial complexes I and IV

Margherita Protasoni[1] (iD), Rafael Pérez-Pérez[2] (iD), Teresa Lobo-Jarne[2] (iD), Michael E Harbour[1], Shujing Ding[1], Ana Peñas[2], Francisca Diaz[3], Carlos T Moraes[3] (iD), Ian M Fearnley[1], Massimo Zeviani[1,4] (iD), Cristina Ugalde[2,5,*] (iD) & Erika Fernández-Vizarra[1,**] (iD)

## Abstract

Mitochondrial respiratory chain (MRC) enzymes associate in super-complexes (SCs) that are structurally interdependent. This may explain why defects in a single component often produce combined enzyme deficiencies in patients. A case in point is the alleged destabilization of complex I in the absence of complex III. To clarify the structural and functional relationships between complexes, we have used comprehensive proteomic, functional, and biogenetical approaches to analyze a MT-CYB-deficient human cell line. We show that the absence of complex III blocks complex I biogenesis by preventing the incorporation of the NADH module rather than decreasing its stability. In addition, complex IV subunits appeared sequestered within complex III subassemblies, leading to defective complex IV assembly as well. Therefore, we propose that complex III is central for MRC maturation and SC formation. Our results challenge the notion that SC biogenesis requires the pre-formation of fully assembled individual complexes. In contrast, they support a cooperative-assembly model in which the main role of complex III in SCs is to provide a structural and functional platform for the completion of overall MRC biogenesis.

**Keywords** complex I; complex III; cytochrome b mutation; mitochondrial respiratory chain assembly; supercomplexes

**Subject Category** Membranes & Trafficking

**The EMBO Journal (2020) 39: e102817**

## Introduction

The mitochondrial respiratory chain (MRC) complex III (cIII) or $bc_1$ complex is a trans-inner-membrane enzyme that couples the transfer of electrons from ubiquinol (reduced coenzyme Q or CoQ) to cytochrome $c$ with the translocation of protons from the mitochondrial matrix to the intermembrane space, by means of the Q-cycle catalytic mechanism (Trumpower, 1990). Biochemically, cIII occupies a central position in the MRC, since it receives electrons from complex I (cI) and complex II (cII) through CoQ and donates them to complex IV (cIV) via cytochrome $c$. CIII is also at the crossroads of alternative electron transfer pathways, such as those from the glycerol-3-phosphate dehydrogenase, electron transfer flavoprotein (ETF), sulfide–quinone reductase (SQR), and dihydroorotate dehydrogenase (DHODH), all converging onto CoQ (Lenaz *et al*, 2007). The quaternary structure of the complex is always dimeric ($cIII_2$), with each monomer being composed of ten different subunits (Iwata *et al*, 1998; Berry *et al*, 1999, 2000), only one of which (MT-CYB) is encoded by the mitochondrial genome (mtDNA). Structurally, $cIII_2$ is part of all known respiratory supercomplexes (SCs), where it physically interacts with both cI and cIV in the SCs $cI+cIII_2+cIV_n$ (Schagger & Pfeiffer, 2000), structures known as "respirasomes" because they are in principle able to transfer electrons from NADH to $O_2$ (Acin-Perez *et al*, 2008; Gu *et al*, 2016; Letts *et al*, 2016; Wu *et al*, 2016; Guo *et al*, 2017). It is well known that severe $cIII_2$ deficiency in patients carrying null mutations in genes encoding some $cIII_2$ structural components and assembly factors are associated with a concomitant decrease in cI activity (Lamantea *et al*, 2002; Bruno *et al*, 2003; Acin-Perez *et al*, 2004; Barel *et al*, 2008; Tucker *et al*, 2013; Carossa *et al*, 2014; Feichtinger *et al*, 2017) and, in some cases, in cIV activity as well (Carossa *et al*, 2014). These pleiotropic effects have been traditionally interpreted as a loss of cI and cIV stability in the absence of their SC partner (Acin-Perez *et al*, 2004),

1   Medical Research Council-Mitochondrial Biology Unit, University of Cambridge, Cambridge, UK
2   Instituto de Investigación Hospital 12 de Octubre (i+12), Madrid, Spain
3   Department of Neurology, Miller School of Medicine, University of Miami, Miami, FL, USA
4   Department of Neurosciences, University of Padova, Padova, Italy
5   Centro de Investigación Biomédica en Red de Enfermedades Raras (CIBERER), U723, Madrid, Spain
    *Corresponding author. Tel: +34 91 7792784; E-mail: cugalde@h12o.es
    **Corresponding author. Tel: +44 1223 252700; E-mail: emfvb2@mrc-mbu.cam.ac.uk

which is based on the premise that the biogenesis of MRC SCs proceeds by the incorporation of pre-made fully assembled individual complexes (Acin-Perez *et al*, 2008).

Here, we have used proteomics and biogenetic approaches to comprehensively characterize the biogenesis and organization of the MRC components in a homoplasmic MT-CYB null mutant human cell line, devoid of fully assembled $cIII_2$ (de Coo *et al*, 1999; Rana *et al*, 2000; Perez-Perez *et al*, 2016). Contrary to the current model (Acin-Perez *et al*, 2004), our data demonstrate that the severe cI deficiency associated with the absence of $cIII_2$ does not originate from destabilization of the fully assembled cI holoenzyme but rather from assembly stalling of nascent cI. MT-CYB mutant cybrid mitochondria accumulate a cI assembly intermediate lacking the catalytic N-module (Mimaki *et al*, 2012; Moreno-Lastres *et al*, 2012; Sanchez-Caballero *et al*, 2016), which is stabilized by the cI assembly factor NDUFAF2 (Ogilvie *et al*, 2005). In addition, we found that specific $cIII_2$ subunits were recruited into stalled protein structures that sequestered cIV subunits, affecting the maturation of the cIV holo-enzyme. These data explain the molecular mechanisms leading to combined respiratory chain deficiency associated with the absence of $cIII_2$, challenge the concept of SC assembly by incorporation of fully assembled individual complexes, and demonstrate the essential role of SCs as $cIII_2$-driven factories carrying out efficient assembly and maturation of the overall mitochondrial respiratory chain.

## Results

### Combined mitochondrial respiratory chain deficiency in Δ4-CYB cybrids

Enzymatic activities of respiratory chain complexes I, II, III, and IV were measured in the #17.3E Δ4-CYB clone, homoplasmic for the 4-bp deletion in *MT-CYB* (heretofore referred to as Δ4-CYB), in comparison with clone #4.1, containing 100% wild-type (heretofore referred to as WT) mitochondrial DNA (mtDNA). Both cybrid clones, obtained from 143B TK⁻ ρ° cells (King & Attadi, 1996a; King & Attardi, 1996b), were populated with mitochondria from the same heteroplasmic patient (Rana *et al*, 2000). In the Δ4-CYB cells, $cIII_2$ activity was virtually absent and the activities of cI and cIV were significantly lower than the WT values, to $25 \pm 13\%$ and $64 \pm 11\%$, respectively (Fig 1A). The profound reduction in cI amounts and activity of Δ4-CYB was confirmed by in-gel activity assays (IGA) and by immunodetection, with a specific antibody against the cI subunit NDUFS1, following blue-native gel electrophoresis (BNGE) separation of the native MRC complexes in mitochondrial extracts from the Δ4-CYB and WT cell lines (Fig 1B and C). The amounts of assembled complexes V and II were not drastically affected by the *MT-CYB* mutation in two different Δ4-CYB clones: #17.3E (E) and #17.3B (B) (Fig 1C).

To investigate the origin of the combined respiratory chain deficiency, we performed stable isotope labeling by amino acids in cell culture (SILAC)-based quantitative proteomics, to compare the relative abundance of proteins from mitochondrial extracts of the two cell lines. Both the Δ4-CYB and the WT cybrids were cultured in "Heavy (H)" and "Light (L)" media and then mixed before mitochondrial isolation by cell disruption, differential centrifugation

(Fernández-Vizarra *et al*, 2010), and solubilization with 1.6 mg n-dodecyl β-D-maltoside (DDM)/mg protein. This experiment was performed in duplicate using reciprocal labeling of the mutant and control cells (Vidoni *et al*, 2017). The resulting fractions were resolved by blue-native gel electrophoresis (BNGE); each lane was then excised in 64 1-mm-thick slices and analyzed by mass spectrometry (MS). This analysis included the relative quantification of 1,263 proteins, the most downregulated of which were structural subunits of cIII and cI (Figs 2A and EV1). The amounts of nine known cI assembly factors (ACAD9, ECSIT, FOXRED1, NDUFAF1, NDUFAF2, NDUFAF3, NDUFAF4, NDUFAF6, and TMEM126B) did not differ significantly between the Δ4-CYB and WT cells (Fig 2A). Conversely, the tenth cI assembly factor detected in this analysis, NDUFAF2 (Ogilvie *et al*, 2005), and the cII assembly factor SDHAF2 (Hao *et al*, 2009) were significantly more abundant in Δ4-CYB mitochondria. Other proteins were also upregulated in the mutant cells. These included CHCHD2, whose knock-down causes cIV deficiency (Baughman *et al*, 2009; Imai *et al*, 2019), HIGD2A, one of the human orthologs of yeast Rcf1, which stabilizes the interaction between $cIII_2$ and cIV (Chen *et al*, 2012; Strogolova *et al*, 2012; Vukotic *et al*, 2012; Rieger *et al*, 2017), and GHITM or growth hormone-inducible transmembrane protein, a member of the BAX inhibitor motif-containing (TMBIM) family. GHITM localizes to the inner mitochondrial membrane, interacts with CHCHD2, and is deemed to participate in the maintenance of mitochondrial morphology and integrity (Oka *et al*, 2008; Meng *et al*, 2017).

To further examine the expression of the structural components of the MRC complexes and the ATP synthase (cV) in the Δ4-CYB cells, we performed metabolic labeling of the thirteen mtDNA-encoded subunits. This analysis indicated that only MT-CYB was not translated in the Δ4-CYB cybrids and that there was no clear reduction in the synthesis of any of the seven ND (cI) subunits (Fig 2B). Next, we tested the steady-state levels of several cI, cII, $cIII_2$, cIV, and cV subunits from whole-cell lysates, separated by SDS–PAGE, and immunovisualized by Western blot (WB) with specific antibodies. $CIII_2$ subunits were in general the most reduced in the Δ4-CYB cybrids, with significantly lower levels of UQCRC1, UQCRC2, UQCRB, UQCRFS1, and UQCRQ (Fig 2C). Conversely, cytochrome c1 (CYC1) showed comparable steady-state levels in both cybrid lines. Likewise, immunodetection using an anti-UQCRQ monoclonal antibody (Abcam ab110255) visualized a band of molecular mass smaller than 10 kDa and revealed equal levels in both cell lines (Fig 2C). Subsequent experiments suggested that this antibody fails to reliably detect UQCRQ, whereas it seems to cross react with UQCR10 (Fig 4C). Several cI subunits also showed variable reduction in their steady-state levels, depending on the structural module with which they associate (Stroud *et al*, 2016; Zhu *et al*, 2016). The lowest levels were detected for NDUFS1, a component of the catalytic N-module, followed by NDUFB8, belonging to the ND5-module, and NDUFS3, belonging to the Q-module. The amounts of NDUFA9 and NDUFB11, assigned to the ND2 and ND4 modules, respectively, were not significantly affected in the mutant cells (Fig 2D). The steady-state levels of mitochondrial and nuclear-encoded cIV subunits were similar to WT cells, except for significantly lower amounts of COX6B1 (Fig 2E), one of the subunits incorporated in the late stages of cIV assembly (Vidoni *et al*, 2017). SDHB of cII was also markedly reduced, to about half in Δ4-CYB cybrids compared with WT

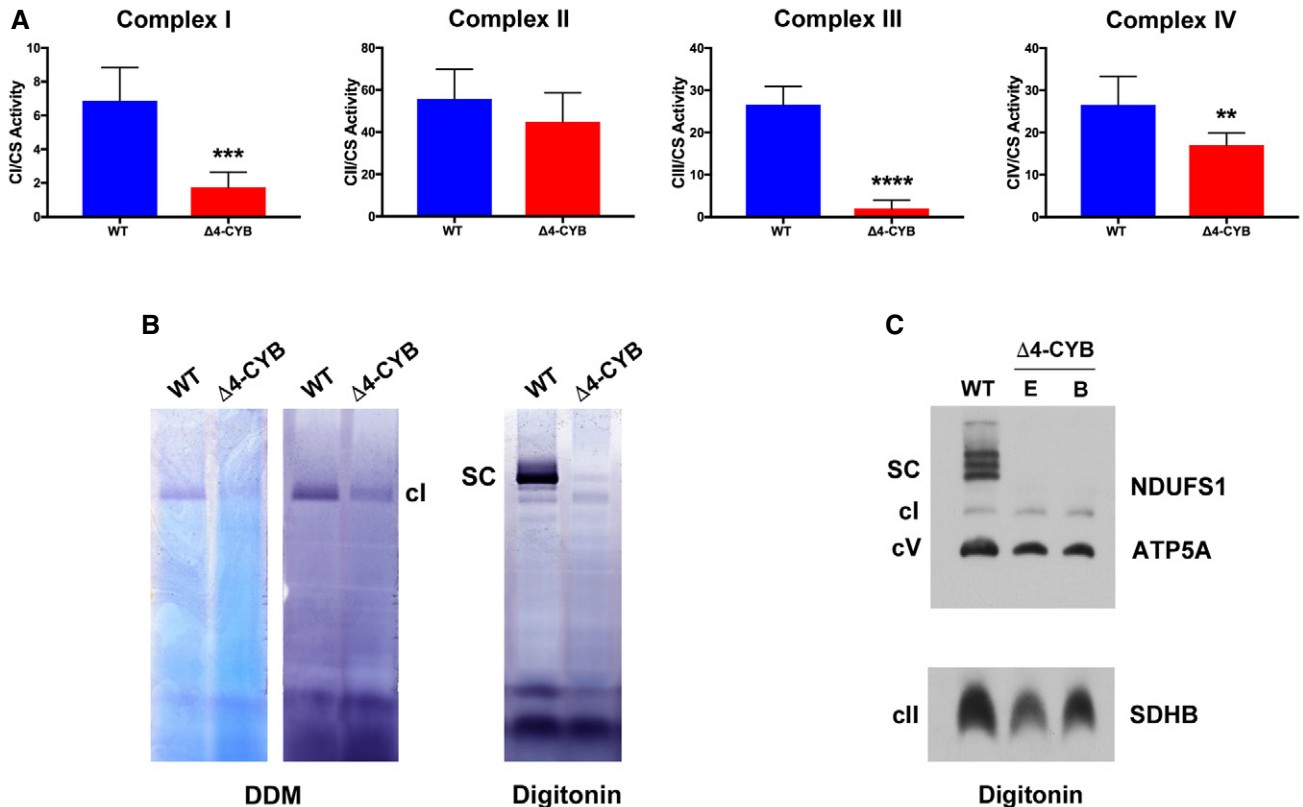

**Figure 1. Complex I and IV enzymatic deficiencies in Δ4-CYB cells.**

A  The activities (mUnits/g of protein) of the MRC enzymes were determined by spectrophotometric kinetic measurements in WT and Δ4-CYB cells and normalized by the percentage of citrate synthase (CS) activity. Results are expressed as mean ± SD (*n* = 4–6 biological replicates). Unpaired Student's *t*-test **P = 0.0100; ***P = 0.0002; ****P < 0.0001.

B  Complex I in-gel activity assays (IGA) after blue-native gel electrophoresis (BNGE) of WT and Δ4-CYB samples solubilized with either 1.6 mg DDM/mg protein or 4 mg digitonin/mg protein. The gels were incubated in the reaction mixture for 1.5 h (lighter signals in DDM gels) or were left to continue the reaction for 24 h to obtain darker signals (DDM and Digitonin gels).

C  BNGE, Western blot, and immunodetection, with anti-NDUFS1 (cI), anti-ATP5A (cV), and anti-SDHB (cII) antibodies, of samples from the WT cybrids and from Δ4-CYB clones E (#17.3E) and B (#17.3B). Clone E was the cell line of choice for the analysis shown in panels (A and B), and all the figures hereafter.

Source data are available online for this figure.

(Fig 2F), whereas cV subunits showed no differences between the two cell lines (Fig 2G).

## The absence of MT-CYB causes accumulation of specific CIII assembly intermediates

Differentially labeled mitochondria solubilized with either 4 mg digitonin (Figs 3 and EV1A and B) or 1.6 mg DDM/mg protein were resolved by BNGE (Fig EV1C and D). Each lane was excised in 64, 1-mm-thick slices, and "complexome profiles" of protein distribution through the gel were obtained by LC/MS analysis (Heide *et al*, 2012; Vidoni *et al*, 2017). By this approach, we compared the relative migration and abundance of the cIII₂ components in Δ4-CYB versus WT clones, in conditions that allow the analysis of individual MRC complexes (DDM) and supercomplexes (SCs, visualized in digitonin-solubilized samples). As expected, fully assembled cIII₂ and cIII₂-containing SCs were missing in the Δ4-CYB cells (Fig EV1). Peptides corresponding to six structural cIII₂ subunits, UQCRC1, MT-CYB, UQCRH, UQCRB, UQCRQ, and UQCR11 and to one

assembly factor, UQCC2, were found only in the datasets corresponding to WT mitochondria (Fig EV1), reflecting again their virtual absence in Δ4-CYB cells. UQCC2 binds to nascent MT-CYB in the very early stages of cIII₂ assembly (Gruschke *et al*, 2012; Tucker *et al*, 2013), and its presence seems to depend on the existence of a functional MT-CYB. In contrast, peptides corresponding to the remaining four subunits, UQCRC2, CYC1, UQCRFS1, and UQCR10; and two assembly factors, BCS1L and MZM1L (LYRM7), were detected in both control and mutated mitochondria (Figs 3A and D, and EV1). To validate the findings obtained from the complexome profiling, we performed BNGE followed by WB and immunodetection analyses of digitonin-solubilized mitochondria. 2D BNGE of Δ4-CYB samples (Fig 3B) confirmed the presence of CYC1 in several protein structures ranging from low to high molecular mass, and the absence of UQCRC2, UQCRFS1, and UQCRQ from cIII₂ structures and from higher molecular size bands visible in Δ4-CYB. Conversely, these subunits were always found in the cIII₂ holocomplex and canonical SCs of WT cells (Fig 3B). Indeed, in the Δ4-CYB cells, residual amounts of UQCRC2, CYC1, and UQCRQ subunits were

immunodetected in a high-molecular size area of 2D-BNGE WB, which does not correspond to that of WT SCs and does not contain UQCRFS1. This protein aggregate was also detected in the top gel slices by MS complexome analyses (Fig EV2).

The relative abundance of each subunit in the assembled species was estimated by calculating the area under the peaks defined by the relative peptide intensity in the complexome profiles obtained from the digitonin-solubilized samples. As "internal standard" experimental controls, we used the amount and distribution within the gel lanes of MRC-unrelated mitochondrial proteins, such as

TOM20, TOM22, and citrate synthase (CS), which were not significantly different between the two cell lines (Fig EV2). These measurements (Fig 3C) confirmed that CYC1 and UQCR10 were more abundant in Δ4-CYB cybrids than UQCRC2 or UQCRFS1, present at extremely low levels and mainly found at the gel migration front (Fig 3A). Conversely, CYC1 and UQCR10 were distributed in several peaks with molecular sizes ranging from 25 to 2,952 kDa, indicating their stabilization inside different assembly intermediates or structural aggregates that accumulate when MT-CYB is lacking and cIII$_2$ biogenesis is impaired. Notably, the amounts and

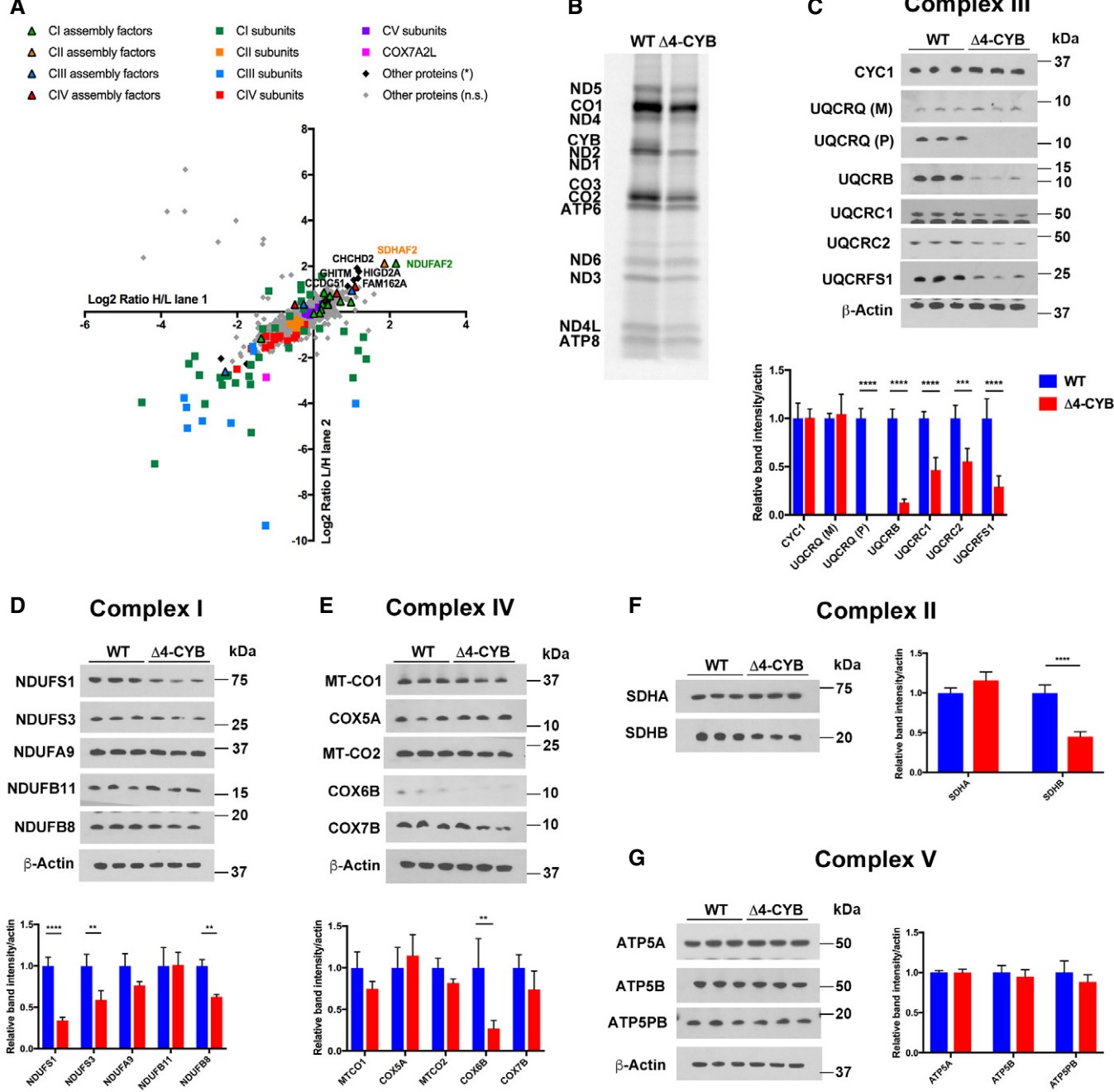

**Figure 2.**

**Figure 2.  Reduced steady-state levels of structural MRC subunits in Δ4-CYB cells.**

A   Scatter plot generated from the peptide content analyzed by mass spectrometry in each of the 64 slices excised from BNGE and after quantifying the heavy-to-light (H/L) and light-to-heavy (L/H) ratios in both reciprocal labeling experiments performed with mitochondria isolated from WT and Δ4-CYB cells (see also Fig EV1). The logarithmic ratios were calculated using MaxQuant (Cox & Mann, 2008), and the statistical significance of the differences for the enrichment or depletion of the proteins was determined with Perseus (Cox & Mann, 2011; Tyanova *et al*, 2016).

B   Labeling of the thirteen mtDNA-encoded MRC structural subunits. Cells were incubated with [$^{35}$S]-L-Met for 1 h in the presence of emetine 100 µg/ml to inhibit cytoplasmic translation.

C   Immunodetection of complex III structural subunits on Western blots of total cell lysates separated by SDS–PAGE, from three independent replicates of WT and Δ4-CYB cells. The graph shows the densitometric quantification of the signals corresponding to each subunit normalized to that of the β-Actin. The mean of the three control (WT) samples was set to 1.0, and all the measurements were referenced to that value. The values plotted in the graphs are the mean ± SD ($n = 3$). Two-way ANOVA with Sidak's multiple comparisons test ****$P < 0.0001$; ***$P = 0.0007$.

D   Immunodetection of complex I structural subunits on Western blots of total cell lysates separated by SDS–PAGE, from three independent replicates of WT and Δ4-CYB cells. The graph shows the densitometric quantification of the signals corresponding to each subunit normalized to that of the β-actin. The mean of the three control (WT) samples was set to 1.0, and all the measurements were referenced to that value. The values plotted in the graphs are the mean ± SD ($n = 3$). Two-way ANOVA with Sidak's multiple comparisons test ****$P < 0.0001$; **$P = 0.0024$ (NDUFS3); **$P = 0.0061$ (NDUFB8).

E   Immunodetection of complex IV structural subunits on Western blots of total cell lysates separated by SDS–PAGE, from three independent replicates of WT and Δ4-CYB cells. The graph shows the densitometric quantification of the signals corresponding to each subunit normalized to that of the β-Actin. The mean of the three control (WT) samples was set to 1.0, and all the measurements were referenced to that value. The values plotted in the graphs are the mean ± SD ($n = 3$). Two-way ANOVA with Sidak's multiple comparisons test **$P = 0.0011$.

F   Immunodetection of complex II structural subunits on Western blots of total cell lysates separated by SDS–PAGE, from three independent replicates of WT and Δ4-CYB cells. The graph shows the densitometric quantification of the signals corresponding to each subunit normalized to that of the β-actin. The mean of the three control (WT) samples was set to 1.0, and all the measurements were referenced to that value. The values plotted in the graphs are the mean ± SD ($n = 3$). Two-way ANOVA with Sidak's multiple comparisons test ****$P < 0.0001$.

G   Immunodetection of complex V structural subunits on Western blots of total cell lysates separated by SDS–PAGE, from three independent replicates of WT and Δ4-CYB cells. The graph shows the densitometric quantification of the signals corresponding to each subunit normalized to that of the β-actin. The mean of the three control (WT) samples was set to 1.0, and all the measurements were referenced to that value. The values plotted in the graphs are the mean ± SD ($n = 3$). There were no differences in the steady-state levels of the tested subunits (2-way ANOVA with Sidak's multiple comparisons test).

Source data are available online for this figure.

distribution of the UQCRFS1-related assembly factors BCS1L (Fernandez-Vizarra *et al*, 2007) and MZM1L (Sanchez *et al*, 2013) were comparable in both cell lines (Fig 3D).

To determine the composition of the UQCR10-containing assembly intermediates by immunopurification, an HA-tagged version of UQCR10 (UQCR10$^{HA}$) was constitutively transduced in WT and Δ4-CYB cells (Fig 4A and B). Its expression was detectable in both cell lines, contrary to UQCRQ$^{HA}$ that was absent in the mutant Δ4-CYB but present in WT. Notably, the anti-UQCRQ antibody cross reacted with UQCR10 (Fig 4C). Both UQCR10$^{HA}$-expressing cybrid lines were grown in "H" and "L" SILAC medium according to a duplicate experimental protocol using reciprocal labeling (Andrews *et al*, 2013) and used for immunopurification of the tagged protein with an anti-HA antibody conjugated to Sepharose beads. MS analysis of immunopurified fractions showed more similar abundance for CYC1 and UQCRH ($\approx-1$ log2 ratio) than for other cIII$_2$ subunits, indicating that the interaction with UQCR10$^{HA}$ was not significantly different in the mutant relative to WT cell lines (Fig 4D). Contrariwise, $< -5$ log2 ratios (32-fold lower) were detected in Δ4-CYB versus WT cells for a number of cIII$_2$ components including UQCRC1, UQCRC2, UQCRFS1, UQCRQ, and UQCRB, in agreement with their very low levels in Δ4-CYB cells. Immunopurified COX7A2L levels, also known as SCAFI (Lapuente-Brun *et al*, 2013), were also much lower in mutant cells, reflecting its preferential binding to cIII$_2$ (Perez-Perez *et al*, 2016). In addition, several cIV structural subunits belonging to the MT-CO2 module (MT-CO2, COX5B, COX6C) or to the MT-CO3 module (COX6B1) were found in the immunopurified fractions, suggesting their physical interaction with UQCR10 in both cell lines, although in lower amount in Δ4-CYB cells (between −2 and −4 in log2 ratio). Three proteins appeared with positive log2 ratios, indicating higher levels of interaction with UQCR10$^{HA}$ in the Δ4-CYB mutants, and therefore are candidates to bind the

UQRC10-containing assembly intermediates found in the mutant cell line. These proteins were as follows: GHITM (Fig 2A); CHCHD3 (MIC19), a member of the mitochondrial contact site (MICOS) complex (Kozjak-Pavlovic, 2017); and HADHB, subunit beta of the fatty acid beta-oxidation trifunctional enzyme (Eaton *et al*, 2000). Upon SILAC-based relative quantification of the total mitochondrial extracts, the only protein showing a change in abundance and distribution associated with the absence of MT-CYB was GHITM (Figs 4E and 2A). However, shRNA-based stable knock-down of either *CHCHD3* or *GHITM* expression did not produce a clear cIII$_2$ assembly or enzymatic defect, ruling out these proteins as *bona fide* cIII$_2$ assembly factors (Fig EV3).

Likewise, CYC1$^{HA}$ expressed in both cell lines (Fig 4F) was perfectly incorporated into cIII$_2$ and SCs in WT cybrids, whereas it presented an atypical CYC1 accumulation pattern in Δ4-CYB cells (Fig 4G), similar to that already seen in the complexome profiles (Fig 3A). These CYC1$^{HA}$ overexpressing cells were then used to perform another duplicate experiment of anti-HA immunopurification combined with SILAC, to compare the interactions of CYC1$^{HA}$ in the two cell lines (Fig 4H). The results further confirmed the low abundance of UQCRC1, UQCRC2, and UQCRFS1 subunits in the mutant cells and again showed a direct interaction of CYC1-containing structures with CIV subunit MT-CO2.

## Incomplete complex I maturation in the absence of MT-CYB

Consistent observations have established that drastic *MT-CYB* mutations are associated with concomitant cIII$_2$ and cI deficiencies (Lamantea *et al*, 2002; Bruno *et al*, 2003; Acin-Perez *et al*, 2004; Carossa *et al*, 2014). This phenomenon was deemed to reflect destabilization of cI after its complete assembly, as reported in a mouse fibroblast cell line carrying a missense mutation in *Mt-Cyb* that

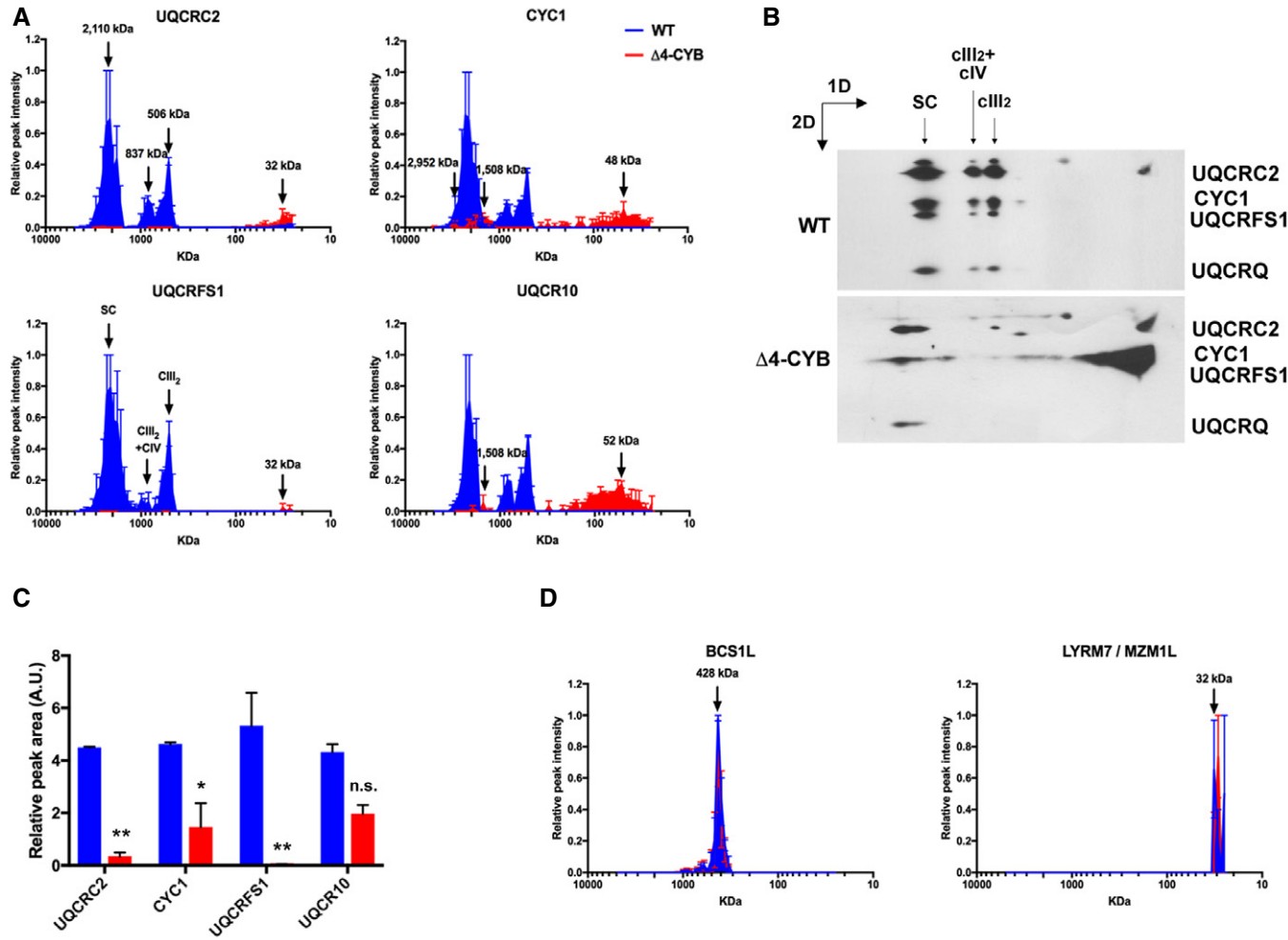

**Figure 3. Blue-native gel electrophoresis (BNGE) mass spectrometry and immunodetection analysis of cIII₂-related proteins.**

A   Complexome profiles of cIII₂ structural subunits generated by analyzing the peptide content in each of the 64 slices in which the gel lanes were excised (see also Figs EV1 and EV2). The graphs plot the relative peptide peak intensities along the lane, setting the maximum to 1.0, versus the molecular mass calculated using the individual complexes and supercomplexes as the standards to generate a calibration curve. The relative amounts of the proteins between the two cell lines were determined by calculating the H/L ratios of peptides that were present in both WT (blue traces) and Δ4-CYB samples (red traces). The represented values are the mean ± SEM of the two reciprocal labeling experiments.

B   Second-dimension BNGE of digitonin-solubilized samples from WT and Δ4-CYB cells, Western blot and immunodetection of the indicated cIII₂ structural subunits with specific antibodies. The immunodetection patterns were equivalent to the complexome profiles.

C   Quantification of the total peak area under the curves (AUC) defined by the peptide intensity peaks for the indicated cIII₂ subunits. The *x*-axis values were the slice number (1-64), and the *y*-axis values were the relative peptide intensity. The graph shows the mean ± SD (*n* = 2). Two-way ANOVA with Sidak's multiple comparisons test **P = 0.0083 (UQCRC2); **P = 0.0033 (UQCRFS1); *P = 0.0224; n.s. = non-significant.

D   Complexome profiles of two cIII₂ assembly factors (BCS1L and LYRM7 or MZM1L) generated in the same way as in (A). The represented values are the mean ± SEM of the two reciprocal labeling experiments.

Source data are available online for this figure.

resulted in the total loss of the protein (Acin-Perez *et al*, 2004). However, our comparative complexome profiling data between human Δ4-CYB homoplasmic cybrids and their isogenic WT controls clearly demonstrate stalling of nascent cI rather than destabilization of the cI holocomplex (Figs 5 and EV4). Peptides corresponding to 31 of the 44 cI structural subunits were detected in both control and mutant samples. To simplify the analysis and data interpretation, the subunits were grouped according to the structural modules in which they are incorporated (Stroud *et al*, 2016; Guerrero-Castillo *et al*, 2017). Quantitative proteomics showed

profoundly decreased amounts of all cI structural subunits in the Δ4-CYB mitochondria (Fig 5A and C). In addition, in the mutant cell line, most of the cI subunits, except those of the N-module, were preferentially accumulating in a peak at an apparent molecular mass of 991 kDa in mitochondria solubilized with digitonin (Fig 5A), or 812 kDa upon solubilization with DDM (Fig EV5A). By contrast, only small amounts (6.1 ± 0.4% of the control) of N-module subunits were found in a peak approximating 1,172 kDa in digitonin-solubilized or 1,002 kDa in DDM-solubilized samples, corresponding to the fully assembled "free" complex I (Figs 5A and D, and

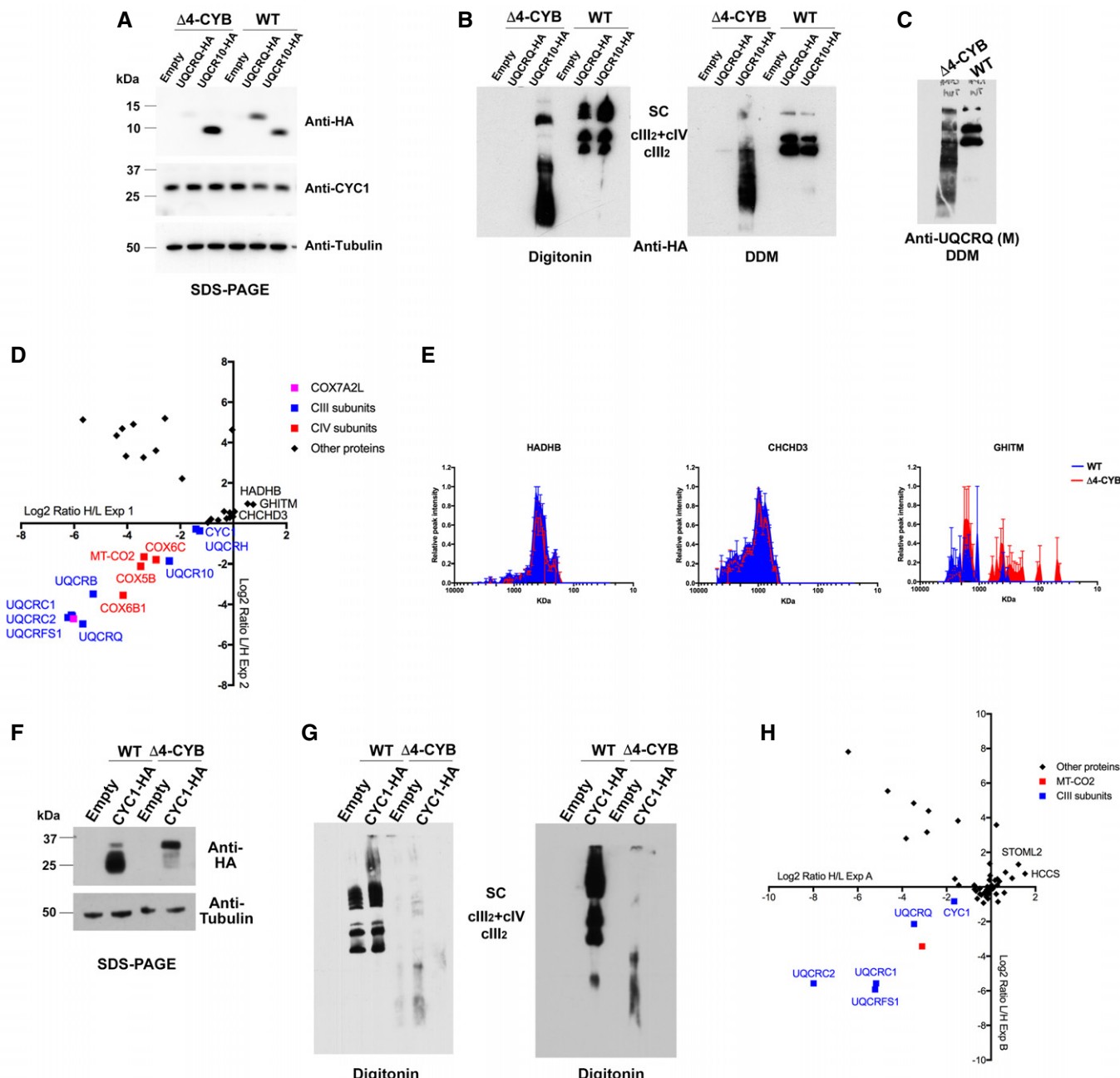

**Figure 4. Proteomic analyses of UQCR10 and CYC1-containing protein associations in Δ4-CYB cells. See also Fig EV3.**

A   SDS–PAGE, Western blot, and immunodetection, with the indicated specific antibodies, of Δ4-CYB and WT cells expressing HA-tagged versions of UQCRQ and UQCR10 and of cells transduced with the lentiviral expression vector without any cDNA insert (Empty).

B   BNGE, Western blot, and immunodetection, with an anti-HA tag antibody, of samples from the same cell lines as in (A) solubilized either with digitonin or DDM.

C   BNGE, Western blot, and immunodetection, with the monoclonal (M) anti-UQCRQ antibody (Abcam ab110255), of non-transduced Δ4-CYB and WT cells. The mitoplast samples were solubilized with DDM (See also Fig EV1).

D   Scatter plot generated from the analysis of the logarithmic heavy (H)-to-light (L) ratios in the *x*-axis and the reverse in the *y*-axis, in the two reciprocal labeling SILAC experiments (1 and 2) and anti-HA immunopurification of Δ4-CYB and WT cells expressing UQCR10[HA].

E   Complexome profiles, generated as in Fig 3, for the proteins found specifically enriched in Δ4-CYB UQCR10[HA], according to the SILAC immunopurification experiments shown in (D). The represented values are the mean ± SEM of the two reciprocal labeling experiments.

F   SDS–PAGE, Western blot, and immunodetection, with the indicated specific antibodies, of Δ4-CYB and WT cells expressing an HA-tagged version of CYC1 and of cells transduced with the lentiviral expression vector without any cDNA insert (Empty).

G   BNGE, Western blot, and immunodetection, with an anti-HA tag antibody, of samples from the same cell lines as in (F) solubilized either with digitonin.

H   Scatter plot generated from the analysis of the logarithmic heavy (H)-to-light (L) ratios in the *x*-axis and the reverse in the *y*-axis, in the two reciprocal labeling SILAC experiments (A and B) of anti-HA immunopurification of Δ4-CYB and WT cells expressing CYC1[HA].

Source data are available online for this figure.

EV5A). This pattern was different from that found in the WT samples solubilized with digitonin, in which all the N-module subunits were exclusively located in the SCs, including the "respirasomes" ($I+III_2+IV_n$), peaking at 2,110 kDa (Figs 5A and EV4). Also, in Δ4-CYB cells, increased amounts of NDUFV1 and NDUFV2 were accumulated at sizes ranging from 25 to 49 kDa, further demonstrating impaired incorporation of the N-module. These profiles indicated that the prominent peak at 991 kDa (digitonin) or 812 kDa (DDM) corresponds to the "pre-complex I" (pre-cI) intermediate that accumulates immediately before the incorporation of the N-module. This is supported by the detection of large amounts of NDUFAF2 in the 991 kDa peak in Δ4-CYB, while it was absent in WT cells (Figs 5B and EV4). NDUFAF2 is a cI assembly factor that stabilizes pre-cI, when the incorporation of the N-module is impaired. The accumulation of this pre-cI species, also known as "~830 kDa intermediate" (Ogilvie *et al*, 2005), was clearly distinct from the 1,002 kDa signal corresponding to the whole cI in both WT and Δ4-CYB mitochondria solubilized in DDM (Fig EV5A). Given their sequence homology, NDUFAF2 probably occupies the binding site of the N-module subunit NDUFA12 in the cI structure, and its upregulation in Δ4-CYB mitochondria could induce cI assembly stalling at the pre-cI stage by preventing the incorporation of the N-module. However, NDUFAF2 overexpression did not prompt the accumulation of pre-cI in the WT cells, and the amount of mature active cI was the same as in the cells transfected with an empty vector (Fig EV5B–D). In addition, downregulation of NDUFAF2 expression had no drastic effects on cI assembly or activity in the WT cells, as previously described (Schlehe *et al*, 2013), and did not promote cI maturation in the Δ4-CYB cells (Fig EV5E–G). These results clearly indicate the occurrence of stalled assembly of cI in the absence of MT-CYB.

To unequivocally determine whether the residual holo-cI observed in the Δ4-CYB cells was due to either degradation of fully assembled holo-enzyme, or assembly stalling of the nascent enzyme, we studied cI assembly dynamics. Pulse-chase [$^{35}$S]-L-Met metabolic labeling was first used to follow the stability and incorporation of the mtDNA-encoded subunits. SDS–PAGE analysis of the individual mitochondrial peptides after a 2-h radioactive pulse and chase times of 2, 5, and 24 h indicated comparable turnover rates of the MT-ND subunits in the presence or absence of MT-CYB (Fig 6A). In addition, 1D-BNGE of these samples ruled out the detection of fully assembled cI and its subsequent degradation over time (Fig 6B). 2D-BNGE analysis of the same samples from WT and Δ4-CYB cells (Fig 6C) showed comparable assembly incorporation of all the mtDNA-encoded subunits in both cell lines at the initial stages (pulse), except for the absence of MT-CYB in the mutant. At this point, the MT-ND subunits accumulated in their respective assembly modules and, to a lesser extent, in the pre-cI. However, after a 2-h chase, while in the WT the MT-ND subunits started to be detected into the SCs, in the Δ4-CYB mutant cells they were exclusively present in the pre-cI spot. At 5- and 24-h chase times, cI assembly progressed directly from pre-cI to SCs in WT, whereas it remained mainly stuck in pre-cI in Δ4-CYB, with very scarce formation of free cI and no trace of SCs.

Then, the incorporation of the cI nuclear-encoded subunits was analyzed at different time points after the recovery of mitochondrial protein synthesis and cI assembly in cells that were pre-treated with the inhibitor doxycycline for 6 days, in order to deplete cells from

MRC mature complexes. 1D-BNGE followed by cI-IGA analysis of samples collected between 0 and 72 h after removal of doxycycline showed the progressive appearance of strong NADH-dehydrogenase activity within the SCs in the WT, whereas in Δ4-CYB cells very faint NADH-dehydrogenase reactive signals were present at a molecular size around 1,000 kDa, corresponding to free cI holocomplex (Fig 6D). IGA analysis also detected faint reactivity in a band of Δ4-CYB with slower electrophoretic mobility, most likely corresponding to a second peak containing the N-module, found in the complexome profile analysis with an apparent molecular size of 1,386 kDa (Fig 5A). Furthermore, 2D-BNGE analysis followed by WB with four different antibodies recognizing specific subunits from distinct cI structural modules (NDUFA9, NDUFB8, NDUFS1, and NDUFV1) was used to determine the cI assembly dynamics on samples collected at the same time points from both cell lines (Fig 6E). In WT cells, the tested cI nuclear subunits were robustly expressed in a position corresponding to SCs with no obvious accumulation of assembly intermediates. By contrast, in Δ4-CYB cells much lower amounts of cI subunits NDUFA9 and NDUFB8 were mostly detected in assembly intermediates as well as in the pre-cI structure. In accordance with the poor cI-IGA signals (Fig 6D), the NDUFS1 and NDUFV1 subunits, belonging to the N-module, were either undetected or detected in two very faint spots corresponding to the cI holocomplex and the higher molecular size structure (asterisk), the complete composition of which must be elucidated. The blots of mutant samples had to be exposed 10 times longer than those of the controls, in order to obtain detectable signals (Fig 6E). These observations further confirmed cI assembly stalling, with very residual incorporation of the N-module and accumulation of pre-cI in the absence of MT-CYB.

## Complex I assembly stalling is due to both structural and functional loss of complex III

To distinguish whether the cI assembly blockage in Δ4-CYB cells is merely due to the structural absence of $cIII_2$ or to the lack of ubiquinol oxidase activity, leading to a CoQ pool redox imbalance, we stably expressed an HA-tagged version of the *Emericella nidulans* alternative oxidase (AOX) (Perales-Clemente *et al*, 2008; Guaras *et al*, 2016) in WT and Δ4-CYB cybrids. AOX is naturally expressed in plants as well as in some fungi and parasites, and its catalytic activity is able to bypass mitochondrial respiratory chain $cIII_2$ and cIV by directly oxidizing CoQ and reducing oxygen (Young *et al*, 2013). AOX overexpression was previously shown to restore CoQ redox balance and to promote cI assembly in *Podospora anserina* fungal strains as well as in mouse cultured cells lacking complex III or IV (Maas *et al*, 2009; Guaras *et al*, 2016). Upon stable transduction of a lentiviral vector containing the AOX[HA] coding sequence, the protein was readily expressed in both WT and Δ4-CYB cybrids (Fig 7A). AOX[HA] was shown to be functional in the human cells because its expression reversed the pyrimidine auxotrophy of the Δ4-CYB cybrids, enabling them to grow in culture medium without uridine (Fig 7B), and indirectly indicating the restoration of the CoQ pool redox balance (King & Attardi, 1989). AOX[HA] expression was associated with a partial recovery of the levels and assembly of cI subunits into the mature enzyme in the Δ4-CYB cells (Fig 7C–E). In agreement, there was a significant change in cI enzymatic activity, increasing from 25 to 55% of the

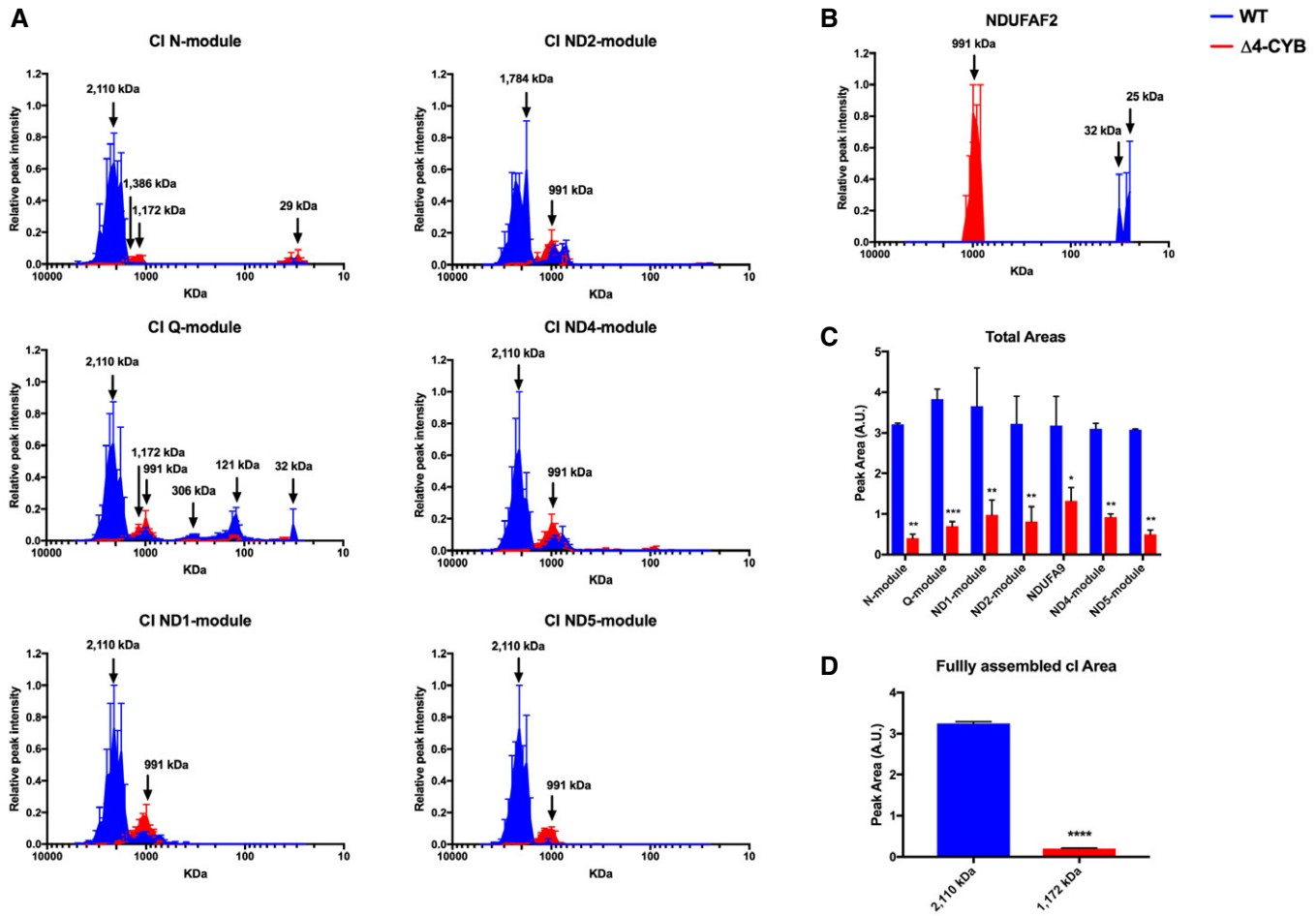

**Figure 5. Complex I alterations in Δ4-CYB cells.**

A   Complexome profiles of the different cI structural modules. The graphs were generated as in Fig 3, but in this case, the peptide intensity values for the individual subunits belonging to each module (Stroud *et al*, 2016) were averaged to simplify the analysis (see also Figs EV4 and EV5). The represented values are the mean ± SEM of the two reciprocal labeling experiments.

B   Complexome profiles of the cI assembly factor NDUFAF2 (see main text for details) in WT (blue) and Δ4-CYB (red) mitochondria (see also Figs EV4 and EV5). The represented values are the mean ± SEM of the two reciprocal labeling experiments.

C   Quantification of the total area under the curve (calculated as in Fig 3C) in the profiles corresponding to each cI module (top graph). The area of the NDUFA9 subunit was calculated separately as its profile did not correspond to any of the other structural subunit modules. The plotted values are the mean ± SD ($n = 2$). Two-way ANOVA with Sidak's multiple comparisons test ***$P = 0.0006$; **$P = 0.0012$ (N-module); **$P = 0.0016$ (ND1-module); **$P = 0.0032$ (ND2-module); **$P = 0.0057$ (ND4-module); **$P = 0.0020$ (ND5-module); *$P = 0.0140$.

D   Quantification of the area of the N-module peak corresponding to the molecular mass of the fully assembled cI: 1,172 kDa in Δ4-CYB samples and the respirasome (cI+cIII$_2$+cIV) SCs in #4.1 WT samples (2,110 kDa). The plotted values are the mean ± SD ($n = 2$). Unpaired Student's *t*-test ****$P < 0.0001$.

WT values (Fig 7F), which was reflected on a maximal respiratory capacity in the Δ4-CYB AOX$^{HA}$ cells of 50% of the WT (Fig 7G). In addition, readily detectable amounts of NDUFAF2 were still bound to pre-cI in the Δ4-CYB AOX$^{HA}$ cells (Fig 7D). Therefore, these results indicate that stable overexpression of functional AOX can only achieve a partial rescue of cI assembly in the absence of cIII$_2$.

To determine whether the impairment of the CoQ redox balance by abolishment of cIII$_2$ activity was sufficient to induce a similar cI assembly defect, WT cells stably expressing AOX$^{HA}$ and their corresponding mock-transduced (EV) controls were cultured for 7 days in the presence of the specific inhibitor antimycin A (AA), which effectively inhibited mitochondrial respiration in the control cells (Fig 7H). Long-term inhibition of cIII$_2$ activity in these cells did not

induce any significant destabilization of cIII$_2$, cI, or SCs (Fig 7I). As expected, WT cells expressing functional AOX$^{HA}$ treated with AA showed normal respiration (Fig 7H), again without affecting the relative distribution and levels of cI and SCs (Fig 7I). These results indicate that adjustment of the CoQ redox state by itself barely influences cI assembly or stability.

**The absence of MT-CYB alters biogenesis of complexes II and IV, but not of complex V**

SILAC–complexome experiments allowed us to systematically analyze the assembly state of the remaining OXPHOS enzymes in Δ4-CYB versus WT cells systematically. Mass spectrometry analysis

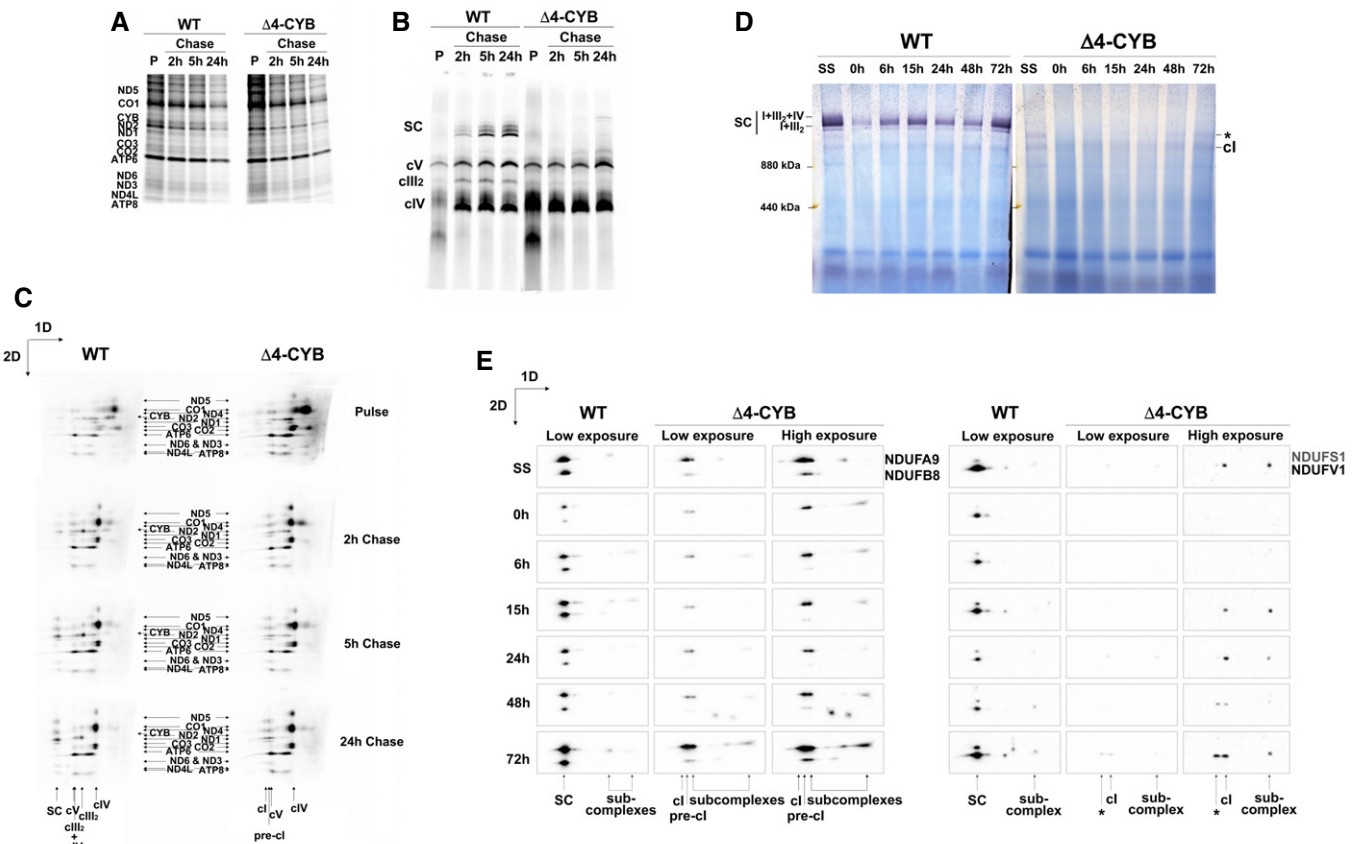

**Figure 6. Complex I assembly kinetics in Δ4-CYB cells.**

A   SDS–PAGE resolving the radioactively labeled mitochondrial translation products after a 2-h pulse (P). The $^{35}$S-Met and the cycloheximide were removed from the medium, and cells were collected at the indicated chase times (2, 5, and 24 h).

B   First-dimension (1D) BNGE analysis of the same cells as in (A), prepared with digitonin.

C   Denaturing second-dimension (2D) BNGE analysis of the same samples allowed following the incorporation of the individual labeled subunits inside their corresponding complex and supercomplex species.

D   Complex I-IGA (cI-IGA) analysis of digitonin-solubilized samples after inhibiting mitochondrial translation with doxycycline for 6 days (0 h). After removing the drug and restoring synthesis of the mtDNA-encoded subunits, the cells were collected at the indicated times to follow the appearance of cI reactivity with time. The gels were incubated in the reaction mixture for 24 h. SS = steady state. The asterisk indicates the presence of a high-molecular-weight cI-containing band of unknown nature (see main text).

E   2D BNGE, Western blot, and immunodetection analysis of WT and Δ4-CYB mitochondria from cells collected at the same times after doxycycline treatment to follow the incorporation kinetics of the indicated cI nuclear-encoded subunits, belonging to different structural modules. The blots shown were either exposed for 16 s. (low exposures) or 160 s. (high exposures) in order to visualize the qualitative signals in the Δ4-CYB samples.

Source data are available online for this figure.

detected subunits of cV belonging to its two distinct assembly modules (He *et al*, 2018). The F1 particle included α (ATP5A1), β (ATP5B), γ (ATP5C1), and ε (ATP5E) subunits, whereas the peripheral stalk included subunits b (ATP5F1), d (ATP5H), F6 (ATP5J), OSCP (ATP5O), e (ATP5I), f (ATP5J2), g (ATP5L), MT-ATP6, MT-ATP8, 6.8PL (MP68), and DAPIT. These results indicate that ATP synthase biogenesis was essentially unaffected in Δ4-CYB (Fig 8A). As for cII subunits, SDHA showed decreased incorporation into the fully assembled enzyme (calculated MW 156 kDa) in the Δ4-CYB cell line, as well as accumulation of sub-assembled species of lower molecular mass. These results could be related to the upregulation of cII assembly factor SDHAF2 (Fig 2A). Accordingly, the amounts of SDHB and SDHC incorporated into cII were about half of the control, and SDHC was also accumulated in subcomplexes (Fig 8B).

However, these alterations in cII biogenesis were not sufficient to determine an overt enzymatic deficiency (Fig 1A) and did not become evident by semi-quantitative Western blot analysis of BNGE samples (Fig 1C).

On the other hand, analysis of the relative amount and distribution of cIV subunits that define each of the assembly modules (Vidoni *et al*, 2017) revealed structural alterations and impaired assembly of cIV in Δ4-CYB cells, compatible with the observed cytochrome *c* oxidase (COX) deficiency (Fig 1A). Subunits from the early, intermediate (MT-CO2), and late (MT-CO3) assembly modules were clearly reduced in free cIV and absent in the positions corresponding to cIII$_2$+cIV and "respirasome" (cI+cIII$_2$+cIV$_n$) SCs. The most affected subunit was NDUFA4 (COXFA4) (Pitceathly & Taanman, 2018), which was shown to have a weaker interaction

with the complex, being incorporated after the assembly of the "canonical" thirteen COX subunits (Balsa *et al*, 2012; Pitceathly *et al*, 2013; Vidoni *et al*, 2017). In addition, the "early" module

composed of COX4I1, COX5A, and HIGD1A appeared to accumulate in subcomplexes in Δ4-CYB mitochondria (Fig 8C). MT-CO1 and COX7B were not detected in this complexome analysis, but BNGE

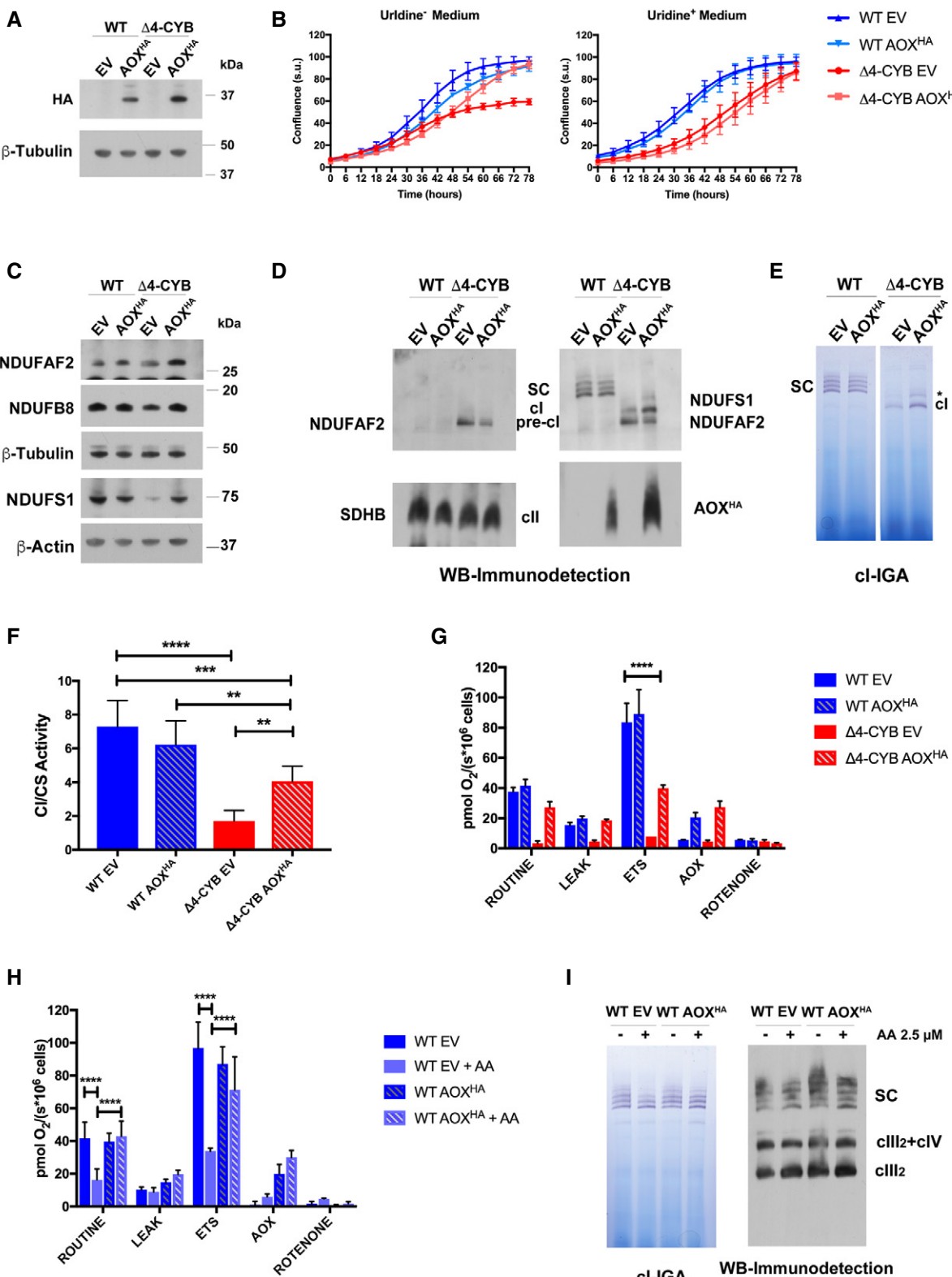

**Figure 7.**

◄

**Figure 7. Alternative oxidase (AOX) expression and function in WT and Δ4-CYB cells.**

A SDS−PAGE, Western blot, and immunodetection of AOX$^{HA}$ expression in whole-cell lysates from WT and Δ4-CYB cells transduced with the AOX$^{HA}$/pWPXLd-ires-Hygro$^{R}$ lentiviral vector. The transduction and selection controls were the same cell lines transfected with an empty pWPXLd-ires-Hygro$^{R}$ vector (EV).

B Growth curves of the AOX$^{HA}$ expressing cell lines and their corresponding EV controls. Cell growth was monitored every 6 h after substituting the medium in two replicate 24-well plates, one plate with medium without uridine (Uridine$^{-}$), and the second plate with medium supplemented with 50 μg/ml uridine (Uridine$^{+}$). The graphs show the average confluence ± SD at each time point ($n$ = 6 wells per cell line).

C Immunodetection of cI structural subunits and NDUFAF2 in the same samples as in panel (A).

D 1D BNGE, Western blot, and immunodetection analyses of digitonin-solubilized mitochondria from WT and Δ4-CYB cybrids expressing AOX$^{HA}$ and their corresponding EV controls.

E Complex I in-gel activity assays (IGA) after BNGE as in panel (D). The gels were incubated in the IGA reaction mixture for 5 h. The asterisk indicates the presence of a high-molecular-weight cI-containing band of unknown nature (see main text).

F Spectrophotometric kinetic measurements of cI activity in WT and Δ4-CYB AOX$^{HA}$ and EV samples normalized by the percentage of citrate synthase (CS) activity. Results are expressed as mean ± SD ($n$ = 5 biological replicates). Two-way ANOVA with Tukey's multiple comparisons test **$P$ = 0.0077 (WT AOX$^{HA}$ versus Δ4-CYB AOX$^{HA}$); **$P$ = 0.0044 (Δ4-CYB EV versus Δ4-CYB AOX$^{HA}$); ***$P$ = 0.0003; ****$P$ < 0.0001.

G High-resolution respirometry analyses performed in intact cells in an Oroboros instrument. ROUTINE: cellular basal oxygen consumption rate (in pmol O$_2$/sec) per million cells in DMEM medium. LEAK is the non-phosphorylating respiration in the presence of the ATP synthase inhibitor oligomycin. ETS: maximal respiration rate in the presence of the uncoupler CCCP. AOX: oxygen consumption in the presence of antimycin A, inhibiting cIII$_2$ activity but not that of AOX. ROTENONE: oxygen consumption in the presence of the cI inhibitor rotenone. In all cases, this was equal to the background. Results are expressed as mean ± SD ($n$ = 2 biological replicates). Two-way ANOVA with Tukey's multiple comparisons test ****$P$ < 0.0001.

H Respirometry analyses, performed as in panel (G), in WT EV controls and AOX$^{HA}$-expressing cells untreated or treated with 2.5 μM antimycin A for 7 days (+AA). Results are expressed as mean ± SD ($n$ = 4 biological replicates). Two-way ANOVA with Tukey's multiple comparisons test ****$P$ < 0.0001.

I 1D BNGE followed by cI-IGA (right) or Western blot and immunodetection of cIII$_2$ subunit CYC1 (left) in digitonin-solubilized samples from WT EV controls and AOX$^{HA}$-expressing cells untreated (−) or treated (+) with 2.5 μM antimycin A for 7 days.

Source data are available online for this figure.

combined with WB and immunodetection with specific antibodies also showed the increased accumulation of these subunits in cIV subassemblies of Δ4-CYB versus WT cell lines (Fig 8D). Moreover, the assembly factor MR-1S was accumulated in the cIV intermediates of mutant samples migrating around the size of mature cIV molecular mass (Vidoni *et al*, 2017) (Fig 8C). The increased binding of the intermediate assembly factor MR-1S, despite the reduced amount of cIV in the Δ4-CYB cells, indicated that the absence of MT-CYB also hampers the maturation and induces stalling of cIV assembly in a series of intermediates.

## Discussion

In order to elucidate the mechanisms controlling the biogenesis of the human respiratory chain in health and disease, we have taken advantage of modern high-throughput proteomics techniques to evaluate the structural re-organization of the MRC complexes in human cybrids characterized by the complete loss of holo-cIII$_2$ (de Coo *et al*, 1999; Rana *et al*, 2000). In addition to the absence of cIII$_2$ enzymatic activity, these cells showed a marked defect in

cI activity. This confirmed the well-established connection between severe cIII$_2$ deficiency and cI impairment. Interestingly, the activity of cIV was also significantly reduced in our system, as previously reported in a subset of patients with severe cIII deficiency (Carossa *et al*, 2014; Feichtinger *et al*, 2017). Accordingly, the absence of fully assembled cIII$_2$ led to the complete loss of SCs containing cIII$_2$ and cI, and to the accumulation of an inactive pre-cI lacking the catalytic N-module; fully assembled cIV levels were also decreased and, to a lesser extent, those of cII as well. Our work gives a straightforward explanation of the combined respiratory chain enzyme deficiency associated with cIII$_2$ depletion, by providing evidence that cIII$_2$-containing SCs are essential to efficiently promote the assembly of the other MRC complexes, particularly of cI.

Previous experimental work in a mouse cell line led to propose a mechanism by which a severe *Mt-Cyb* mutation determined cI instability and degradation once the cI holo-enzyme was fully formed (Acin-Perez *et al*, 2004). The origin of this instability was deemed to be the oxidative damage to cI protein components due to reverse electron transfer (RET) originated by high reduced CoQ to total CoQ (CoQH$_2$/CoQ) ratios, triggering a response to degrade damaged cI

**Figure 8. Assembly state of cV, cII, and cIV in Δ4-CYB cells.**

A Complexome profiles of the two cV structural and assembly modules. The graphs were generated as in Fig 3, but in this case, the peptide intensity values for the individual subunits belonging to each module (He *et al*, 2018) were averaged to simplify the analysis. The represented values are the mean ± SEM of the two reciprocal labeling experiments.

B Complexome profiles of the three detected cII subunits. The graphs were generated as in Fig 3. The represented values are the mean ± SEM of the two reciprocal labeling experiments.

C Complexome profiles of the different cIV assembly modules and of the last subunit to be incorporated (NDUFA4). The graphs were generated as in Fig 3, but in this case, the peptide intensity values for the individual subunits belonging to each assembly module (Vidoni *et al*, 2017) were averaged to simplify the analysis. The represented values are the mean ± SEM of the two reciprocal labeling experiments. The bar graph represents the quantification of the total area under the curve (calculated as in Fig 3C) in the profiles corresponding to each cIV module. The plotted values are mean ± SD ($n$ = 2). Two-way ANOVA with Sidak's multiple comparisons test **$P$ = 0.0088 (Early); **$P$ = 0.0063 (MT-CO2); **$P$ = 0.0040 (NDUFA4); *$P$ = 0.0264.

D 1D BNGE, Western blot, and immunodetection of two cIV subunits (MT-CO1 and COX7B) in samples from WT and Δ4-CYB cells solubilized with DDM and digitonin (Dig).

Source data are available online for this figure.

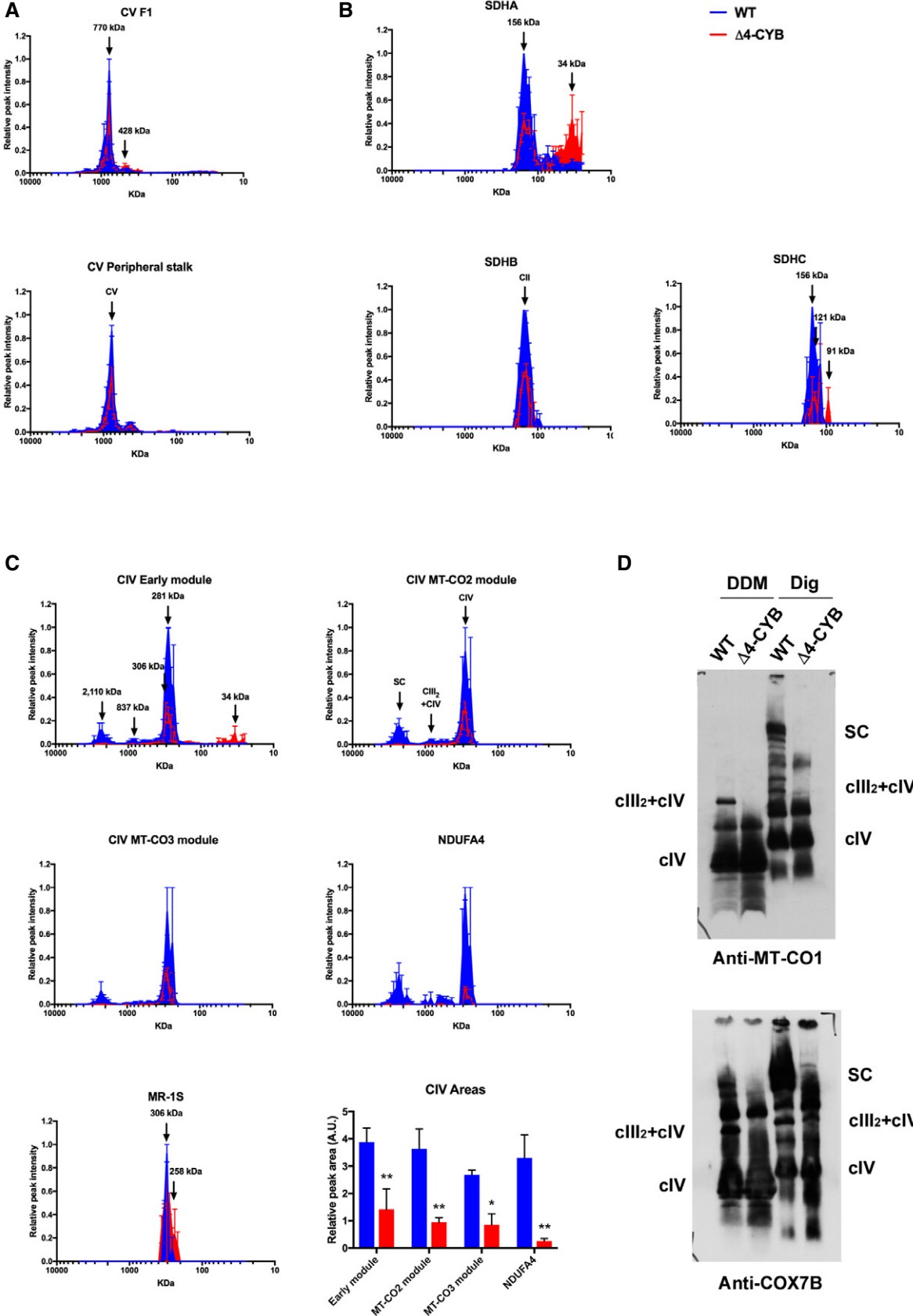

**Figure 8.**

by undefined mechanisms and proteases (Guaras *et al*, 2016). However, this explanation is incompatible with the fact that long-term exposure to $cIII_2$ inhibitors, which would also increase the $CoQH_2/CoQ$ ratio, does not result in cI destabilization (Acin-Perez *et al*, 2004; Guaras *et al*, 2016; this work).

The results shown in this report clearly prove that the absence of $cIII_2$ originates a block in the assembly process of nascent cI, demonstrating that the efficient maturation of human cI strictly depends on its association to $cIII_2$ within SCs in physiological conditions. Our conclusion is based on the following evidence. First, NDUFAF2, the factor that holds pre-cI (or ~830 kDa intermediate) in a competent state to complete the assembly process by adding the N-module (Ogilvie *et al*, 2005; Sanchez-Caballero *et al*, 2016), is overexpressed in Δ4-CYB mitochondria as a consequence of cI assembly stalling and was detected bound to immature pre-cI. Second, pulse-chase labeling of the mtDNA-encoded subunits showed comparable turnover of the MT-ND subunits between the mutated and control cells, contrary to the model based on instability and degradation of fully assembled cI. Third, the kinetics of incorporation of both mitochondrial and nuclear cI subunits into the nascent complex over time indicated the accumulation of most cI subunits in the pre-cI at all times in the mutant, rather than the prior formation of a complete "free" cI able to be incorporated into SCs and then degraded in the absence of $cIII_2$. Several arguments may explain the different experimental interpretation in mouse cells carrying a *Mt-Cyb* mutation (Acin-Perez *et al*, 2004). In addition to using cells from different species, in which the turnover rates of the MRC complexes may differ, the times used in the pulse-chase experiments of Acin-Perez *et al* missed intermediate points (e.g., 2 and 5 h), and samples were solubilized using DDM instead of digitonin, which prevents the visualization of respirasomes. Indeed, Acin-Perez *et al* describe the appearance of a second faster-migrating cI band of "unknown nature" accumulating in the *Mt-Cyb*-mutated cells, that could very well be pre-cI, which was still unknown at the time (Ogilvie *et al*, 2005). Importantly, our detailed analysis of respiratory chain complex subunit incorporation dynamics supports the view that full and proper maturation of human cI preferentially occurs within the SC structures (Moreno-Lastres *et al*, 2012). By increasing the intermediate time points of our experiments, we demonstrated that the temporal gap between the complete assembly of cI and its incorporation into SCs, as previously proposed (Acin-Perez *et al*, 2008; Guerrero-Castillo *et al*, 2017), did not occur. Notably, the existence of such a kinetic "gap" had previously set the basis to propose the "plasticity model" for the structural and functional organization of the respiratory chain, which, in view of our data, deserves further confirmation.

The replacement of $cIII_2$ function by xeno-expression of AOX in Δ4-CYB cells partially increased the levels of functional cI, as reported in mutant mouse cells (Guaras *et al*, 2016). However, pre-cI accumulation was still evident, and cI-linked activities remained at approximately half of the control values. These observations are consistent with the idea that promotion of efficient cI maturation has two components: a functional component sensing the redox state of the CoQ pool and a structural component induced by the physical binding of $cIII_2$ to pre-cI. Concerning the CoQ redox balance, one could speculate that a highly reduced CoQ pool can change the redox state of pre-cI within which the Fe-S clusters would be reduced, thus possibly changing the milieu polarity and

impeding the incorporation of the N-module through electrostatic forces. From the structural point of view, there is no direct contact between $cIII_2$ and the N-module of cI in the respirasome (Gu *et al*, 2016; Letts *et al*, 2016; Wu *et al*, 2016; Guo *et al*, 2017), but it has now been demonstrated that there is a structural cross-talk between cI and $cIII_2$ (Letts *et al*, 2019). Therefore, it is conceivable that the binding of $cIII_2$ to pre-cI could induce a conformational change facilitating the exchange of NDUFAF2 for the N-module.

Our analyses also showed stabilization of novel $cIII_2$ intermediates containing catalytic CYC1 and accessory UQCR10 subunits. This finding was surprising since MT-CYB, the only mtDNA-encoded subunit of $cIII_2$, has consistently been considered as the "seed" around which the rest of $cIII_2$ builds up, assuming that no intermediates could be assembled independently from MT-CYB (Ndi *et al*, 2018). Moreover, the current $cIII_2$ assembly model establishes an early interaction of CYC1 with UQCRC1 and UQCRC2 (Zara *et al*, 2009; Signes & Fernandez-Vizarra, 2018). Our data are not compatible with this hypothesis since CYC1 accumulated in different protein structures, whereas UQCRC1 and UQCRC2, whose stability critically depends on the presence of MT-CYB, were virtually undetected. Although we found UQCR10- and CYC1-containing protein structures within a wide range of molecular sizes, mass spectrometry analysis on immunopurified fractions overexpressing $UQCR10^{HA}$ and $CYC1^{HA}$ failed to detect convincing novel partners potentially acting as $cIII_2$ chaperones or assembly factors. By contrast, these analyses provided evidence for the co-existence of cIV structural subunits, mainly belonging to the MT-CO2 assembly module (Vidoni *et al*, 2017), within the $cIII_2$ stalled intermediates present in the *MT-CYB* mutant cells, where SCs are totally absent. Overall, these findings suggest that cIV subunits are sequestered within aberrant $cIII_2$ subcomplexes, possibly as a signal to avoid the full assembly and maturation of cIV in a cellular environment lacking SCs. Accordingly, the overall biogenesis of cIV was affected by the loss of $cIII_2$ without enhanced turnover of cIV subunits and holo-cIV. This suggests stalling of cIV maturation, further supported by the increased amounts of MR-1S, a mammalian COX assembly factor found within the advanced cIV intermediates (Vidoni *et al*, 2017). Likewise, our analyses showed general assembly adaptations of cII in the mutant cells, supported by overexpression of SDHAF2, a cII assembly factor binding SDHA and mediating its flavinylation (Hao *et al*, 2009; Kim *et al*, 2012). However, these alterations were not drastic enough to induce a significant functional impairment of cII.

In conclusion, our data have uncovered a fundamental function for MRC SCs, as the platform for regulated assembly of complexes I and IV, and possibly cII as well, in which $cIII_2$ is central. Thus, SC could act as "mitochondrial factories" to control the stoichiometry of the MRC complexes in different metabolic settings or even assist in the overall repair of respiratory chain modules damaged during respiratory work. Both possibilities will be addressed in future investigations.

# Materials and Methods

### Cell culture

143B-TK⁻ osteosarcoma-derived cybrid cells were grown in High-Glucose (4.5 g/l) DMEM containing Sodium Pyruvate and

GlutaMAX™ (Gibco-Thermo Fisher Scientific) supplemented with 10% fetal bovine serum (Gibco-Thermo Fisher Scientific) and 50 µg/ml uridine (Sigma-Aldrich). Cells transduced with expression vectors containing a puromycin resistance cassette were grown in the presence of 1 µg/ml Puromycin (InvivoGen). If it was hygromycin, the final concentration of the Hygromycin B was 100 µg/ml (InvivoGen), and for neomycin-resistant cells, the final concentration of geneticin was 500 µg/ml (Gibco-Thermo Fisher Scientific).

The cells used in the SILAC experiments were grown in DMEM for SILAC (Gibco-Thermo Fisher Scientific) plus 10% dialyzed serum (Gibco-Thermo Fisher Scientific) and 50 µg/ml uridine, supplemented either with unlabeled L-lysine monohydrochloride ($K_0$), L-arginine ($R_0$), and L-proline ("Light" conditions) or with L-lysine-$^{13}C_6$,$^{15}N_2$ hydrochloride ($K_8$), L-arginine-$^{13}C_6$,$^{15}N_4$ hydrochloride ($R_{10}$), and L-proline ("Heavy" conditions), all from Sigma-Aldrich.

## Denaturing and native electrophoresis, western blot, and immunodetection

Total protein extracts from cultured cells or fractions were resolved by SDS–PAGE using Novex NuPAGE 4–12% Bis-Tris Precast Gels (Thermo Fisher Scientific).

Samples for blue-native gel electrophoresis (BNGE) were prepared either from digitonized cellular extracts (Nijtmans *et al*, 2002) or from isolated mitochondria (Fernández-Vizarra *et al*, 2010). For the solubilizations, either 1.6 mg DDM/mg protein or 4 mg digitonin/mg protein was used (Wittig *et al*, 2006; Acin-Perez *et al*, 2008). Approximately 50 µg of protein was loaded into Native PAGE Novex 3–12% Bis-Tris Protein Gels (Thermo Fisher Scientific) and electrophoresed in the conditions indicated by the manufacturer.

Proteins were electroblotted to PVDF membranes and immunodetected using commercial antibodies (Table 1). Immunoreactive bands were visualized using ECL Western Blotting Detection Reagents (GE Healthcare) and X-Ray films (Fuji) or using a digital Amersham Imager 680 (GE Healthcare). Signal intensities were quantified by densitometry using ImageJ.

## Enzymatic activity assays

For the biochemical kinetic reaction assays, digitonin-solubilized cell samples were used (Tiranti *et al*, 1995). Individual cI (NADH: decylubiquinone oxidoreductase, rotenone sensitive), cII (succinate: $CoQ_1$ oxidoreductase), cIII (decylubiquinol:cytochrome c oxidoreductase), cIV (cytochrome *c* oxidase), and citrate synthase (CS) activities were measured as described (Kirby *et al*, 2007), with slight modifications. The reactions were performed for 2 min in 96-well plates in a final volume of 200 µl.

Complex I in-gel activity (IGA) assays were performed on the samples electrophoresed in native conditions as described (Calvaruso *et al*, 2008). To allow for cI activity to appear, gels were incubated between 1.5 and 24 h in cI-IGA reaction buffer.

## Complexome profiling

Mitochondria isolated from 1:1 mixtures of differentially SILAC labeled #4.1 WT and #17.1 Δ4-CYB cybrids were used to analyze

**Table 1. Antibodies used in this study.**

| Antibody | Source | Catalogue number |
|---|---|---|
| Rabbit monoclonal anti-NDUFS1 | Abcam | ab169540 |
| Mouse monoclonal anti-NDUFS3 | Abcam | ab110246 |
| Mouse monoclonal anti-NDUFA9 | Life Technologies | LS459100 |
| Mouse monoclonal anti-NDUFB11 | Proteintech | 16720-1-AP |
| Mouse monoclonal anti-NDUFB8 | Abcam | ab110242 |
| Rabbit polyclonal anti-NDUFAF2 | Proteintech | 13891-1-AP |
| Mouse monoclonal anti-SDHA | Abcam | ab14715 |
| Mouse monoclonal anti-SDHB | Abcam | ab14714 |
| Mouse monoclonal anti-UQCRC1 | Abcam | ab110252 |
| Mouse monoclonal anti-UQCRC2 | Abcam | ab14745 |
| Mouse monoclonal anti-UQCRFS1 | Abcam | ab14746 |
| Rabbit polyclonal anti-CYC1 | Proteintech | 10242-1-AP |
| Mouse monoclonal anti-UQCRQ | Abcam | ab110255 |
| Rabbit polyclonal anti-UQCRQ | Abcam | ab136679 |
| Rabbit monoclonal anti-UQCRB | Abcam | ab190360 |
| Mouse monoclonal anti-COX5A | Abcam | ab110262 |
| Mouse monoclonal anti-COX6B1 | Abcam | ab110266 |
| Mouse monoclonal anti-COX7B | Abcam | ab197379 |
| Mouse monoclonal anti-MTCO1 | Abcam | ab14705 |
| Mouse monoclonal anti-MTCO2 | Abcam | ab110258 |
| Mouse monoclonal anti-ATP5A | Abcam | ab14748 |
| Rabbit polyclonal anti-ATPbeta | Santa Cruz Biotechnology | SC-33618 |
| Mouse monoclonal anti-ATPb | Santa Cruz Biotechnology | SC-514419 |
| Mouse monoclonal anti-β-actin | Sigma-Aldrich | AI978 |
| Mouse monoclonal anti-β-tubulin | Sigma-Aldrich | T5201 |
| Rat monoclonal anti-HA | Roche | 11867423001 |
| Mouse monoclonal Anti-FLAG M2 | Sigma-Aldrich | F3165 |
| Goat anti-Mouse IgG | PROMEGA | W402B |
| Goat anti-Rabbit IgG | PROMEGA | W401B |
| Goat anti-Rat IgG | Cell Signaling Technology | 70775 |
| Rabbit monoclonal anti-HA (Sepharose® Bead Conjugate) | Cell Signaling Technology | 3956S |

the samples and generate the complexome profiles as described (Vidoni *et al*, 2017).

## Overexpression of tagged proteins

For the amplification of UQCRQ and UQCR10 cDNAs, total RNA was extracted from control human fibroblasts (TRIzol Plus RNA Purification System, Invitrogen) and retrotranscribed (Omniscript RT Kit, Qiagen). Approximately 200 ng of cDNA were used as templates for the amplification of UQCRQ and UQCR10 using specific primers (sequences available upon request). The PCR products were cloned

directly into the pCR2.1 TA-cloning vector. After sequence verification, selected clones were used to add the C-terminal HA tags by PCR amplification. The CYC1[HA] PCR product was generated using DNA from synthetic clone containing the CYC1 sequence as the template. The HA-tagged *Emericella nidulans* AOX cDNA insert cloned into the pTNT vector was a kind gift from Mª Pilar Bayona-Bafaluy (University of Zaragoza, Spain). The insert was amplified by PCR with specific oligos using this vector as a template. The PCR-generated HA-tagged fragments were again cloned in the pCR2.1 TA-cloning vector (Invitrogen). These inserts were then introduced into the pWPXLd-ires-Puro[R] or pWPXLd-ires-Hygro[R] lentiviral expression vectors, modified versions of pWPXLd (Addgene #12258), by restriction enzyme digestion with EcoRV/PmeI and BamHI and ligation with T4 DNA ligase (New England Biolabs).

Lentiviral particles were generated in HEK293T packaging cells by co-transfection, with FuGENE 6 (Promega), of the target vector with the packaging psPAX2 (Addgene plasmid #12260) and envelope pMD2.G (Addgene #12259) vectors. The lentiviral vectors were a gift from Didier Trono. Target cells were transduced as described (Perales-Clemente *et al*, 2008). Twenty-four hours after transduction, cells were selected for puromycin or hygromycin resistance.

The Myc-DDK (FLAG)-tagged NDUFAF2 cDNA clone (Cat# RC207387) and the "empty" pCMV6-Entry mammalian expression vector (Cat# PS100001) were purchased from OriGene Technologies. WT and Δ4-CYB cybrids were transfected with these vectors using FuGENE HD (Promega). Forty-eight hours after transfection, cells were selected for neomycin resistance.

### Immunopurification and quantitative proteomics

The immunopurification of the HA-tagged cIII$_2$ subunits UQCR10 and CYC1 expressed in the SILAC labeled WT and Δ4-CYB cybrids was performed and analyzed as described (Andrews *et al*, 2013; Vidoni *et al*, 2017).

### Assembly kinetics assays

[35S]-L-Methionine pulse and pulse-chase labeling of the mitochondrial peptides were performed as described (Chomyn, 1996; Fernandez-Silva *et al*, 2007). Labeled samples from cells in one 10-cm petri dish were prepared for BNGE using the digitonization method (Nijtmans *et al*, 2002).

Mitochondrial translation was blocked by adding 15 μg/ml doxycycline to the cell culture medium for 6 days, and the cells were collected at different time points after removal of the drug and prepared for BNGE as described (Ugalde *et al*, 2004).

### Cell growth

Growth curves were assessed using an IncuCyte HD instrument (Essen Bioscience) using an algorithm to calculate cell confluency based on inverted microscope imaging. Images of each of the 24 wells in the plates were taken every 6 h for a total period of 78 h.

### Respirometry

Oxygen consumption measurements were performed in intact cells using an Oroboros Instruments High-Resolution Respirometer (Pesta & Gnaiger, 2012). Measurements were performed using approximately $5 \times 10^6$ cells in the same supplemented culture medium used to grow them. Basal (ROUTINE) respiration was recorded until the steady state was reached and was continued for an additional 5 min. To inhibit the ATP synthase (cV) and measure the non-phosphorylating respiration (LEAK), 2.5 μM oligomycin was added to the chambers and the respiration rates left to reach the steady state. The uncoupled state (ETS) was achieved by titrating CCCP in 0.5 μM steps until the respiratory rates did not increase any further. Next, 2.5 μM antimycin A was added to the chambers to inhibit cIII and to evaluate AOX-driven respiration. Finally, 1 μM rotenone was added to inhibit cI.

### Statistical analysis

Numerical data in graphs are shown as mean ± SD. Student's *t*-test was used for pair-wise comparison, while 2-way ANOVA with Sidak's or Tukey's (depending on the software recommendation) *post hoc* test was used for multiple comparisons. GraphPad Prism v.7.0e for Mac OS was the software used for the statistical analyses and for calculation of the area under the curves (AUC) defined by the peptide intensity in each gel slice in the MS complexome profiling analyses.

## Data availability

The mass spectrometry proteomics data have been deposited to the ProteomeXchange Consortium (https://www.ebi.ac.uk/pride/) via the PRIDE partner repository with the dataset identifier PXD016521.

**Expanded View** for this article is available online.

### Acknowledgements

We are grateful to Antonio Barrientos (University of Miami, USA) for critically revising the manuscript and to Justin G. Fedor (Mitochondrial Biology Unit, Cambridge, UK), Leonid A. Sazanov (IST Austria, Klosterneuburg, Austria), and James A. Letts (University of California, Davis, USA) for scientific discussions. We also thank Irenaeus F. de Coo (Erasmus University, The Netherlands) for the original patient fibroblasts used to produce cybrids and Ester Perales-Clemente and M. Pilar Bayona-Bafaluy (University of Zaragoza, Spain) for providing the Puro[R] and Hygro[R] lentiviral expression vectors. Our research was supported by the Core Grant from the Medical Research Council (Grant MC_UU_00015/5), ERC Advanced Grant FP7-3222424 and NRJ-Institut de France Grant (to M.Z.), Association Française contre les Myopathies (AFM) grant 16086 (to E.F-V.), Instituto de Salud Carlos III-MINECO/European ERDF-ESF grant PI17-00048, Comunidad Autónoma de Madrid/ERDF-ESF grant P2018/BAA-4403, and NIH-RO1 grant GM105781 (to C.U.).

### Author contributions

Conceptualization, EF-V and CU; Methodology, MEH, IMF, CU, and EF-V; Software, MEH; Investigation, MP, RP-P, TL, MEH, SD, AP, and EF-V; Formal Analysis, MP, RP-P, MEH, SD, CU, and EF-V; Resources, FD, CTM, MZ, CU, and EF-V; Writing—Original Draft, EF-V; Writing—Reviewing and Editing; MEH, FD, CTM, IMF, MZ, CU, and EF-V; Visualization, MP, RP-P, CU, and EF-V; Supervision, CU and EF-V; Funding Acquisition, CU, EF-V, and MZ.

## Conflict of interest

The authors declare that they have no conflict of interest.

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
