## [Review Process File · The EMBO Journal]

Respiratory supercomplexes act as a platform for complex III-mediated maturation of human mitochondrial complexes I and IV

Margherita Protasoni, Rafael Pérez-Pérez, Teresa Lobo-Jarne, Michael E. Harbour, Shujing Ding, Ana Peñas, Francisca Diaz, Carlos T. Moraes, Ian M. Fearnley, Massimo Zeviani, Cristina Ugalde and Erika Fernández-Vizarra.

Review timeline:

Submission date:	29 th June 2019
Editorial Decision:	24 th July 2019
Revision received:	17 th October 2019
Editorial Decision:	29 th October 2019
Revision received:	2 nd November 2019
Accepted:	26 th November 2019

Editor: Elisabetta Argenzio

Transaction Report:

1st Editorial Decision

24th July 2019

Thank you for submitting your manuscript entitled "Respiratory supercomplexes as the platform for complex III-mediated maturation of complexes I and IV" to The EMBO Journal. Your study has been sent to three referees for evaluation, whose reports are enclosed below.

As you can see, while the reviewers find the manuscript potentially interesting, they also raise four key points that need to be addressed before they can support publication in The EMBO Journal. In particular, referee #1 and #2 request you to further investigate i) the role of the assembly factor NDUFAF2, ii) the effect of coenzyme Q oxidation by alternative oxidases, iii) the effects that loss of complex III has on complex I, and iv) mutations in other complexes.

Given the overall interest of your study, I would like to invite you to revise the manuscript in response to the referee reports. I should also note that conclusively addressing these issues and all the minor points raised by the referees is essential for publication in The EMBO Journal.

REFeree REPORTS

Referee #1:

This work provides evidence for interplay between the redox complexes in terms of biogenesis.

I only have one point of criticism. In the beginning of the Introduction:

"..with the translocation of four protons from the mitochondrial matrix to the intermembrane space."

referring to complex III. This is erroneous and must be corrected.

Referee #2:

In this manuscript, M. Protosoni and co-workers study a cell line with a mutation in MT-CYB that results in deficiency of the mitochondrial respiratory chain complex III. Deficiencies in complex III have been previously reported to cause complex I deficiencies and the reported cell line shows a specifically strong reduction in the latter enzyme. It was previously assumed that this interdependency is indirect and that complex I decrease is caused by a destabilization of the fully assembled enzyme. In contrast, the current work shows that it is instead the failure to complete the assembly at a late stage. This late stage occurs by the addition of the NAD dehydrogenase module (N-module), which according to the here-proposed model needs to occur in an association with complex III in a supercomplex. If true, this model would resolve long-standing conundrums relating to the independency of respiratory chain complex assembly and the role(s) of respiratory supercomplex formation. Overall, the manuscript clearly describes a set of high quality data obtained from the study of the D4-CYB cell line. It is, however, clear from the presented data that this mutation has very severe consequences for mitochondrial fitness and impacts on all respiratory chain enzymes. In order to substantiate the proposed model, the authors should address experimentally two major points:

1. A key finding of this manuscript is that the assembly factor NDUFAF2 is highly upregulated in the D4-CYB cell line and binds to the late complex I assembly intermediate. Because NDUFAF2 occupies a position similar to the N-module, it is possible that the molecular explanation for the inability to complete complex I assembly are the increased levels of the assembly factor in the D4-CYB cell line. This could impair progression of assembly by blocking (parts of) the N-module binding sites. Such a hypothesis would be in line with data showing that complex I assembly can progress in absence of NDUFAF2 (PMID: 23702311). Hence, the authors should (1) deplete NDUFAF2 from the D4-CYB cell line by siRNA or similar approaches, and (2) overexpress NDUFAF2 in wild type cells, and test whether this either restores or impairs assembly, respectively.
2. In fungi assembly of complex I is not inhibited by mutations abolishing supercomplex formation due to absence of complex III and/or complex IV (PMID: 19111556 and references therein). A key feature of these fungi is that they can upregulate the expression of alternative oxidase (AOX), which directly oxidises reduced coenzyme Q, thereby alleviating the oxidative stress load of mitochondria. Consequently, it is possible that the molecular reason why complex III deficiency does not provoke loss of complex I activity is the presence of AOX in these fungi. Therefore, the authors should express AOX in the D4-CYB cell line to test whether reducing the accumulation of reduced coenzyme Q corrects complex I activity in a similar fashion. Expressing AOX would also allow correcting the respiratory deficiency of the D4-CYB cell line and comparing these cells better to the respiratory competent wild type cells.

Minor points:

1. Complex II should be analysed also by BN-PAGE with western blotting. While the activity measurements do not indicate decreased levels, the complexome and steady state data show that also complex II is reduced. This is an important experiment because complex II, which is not part of the supercomplexes, could represent a good control for the specificity of the assembly defects.
2. The authors should expand their mechanistic discussions in regards to the recent structural work. Can they speculate further how the binding of complex III to complex I facilitates the finalization of assembly?

Referee #3:

In this manuscript, Protosoni et al studied the assembly of respiratory complexes in a mutant cell line deficient in MtCYB, a subunit of complex III encoded by mitochondrial DNA. As a result of

CYB deficiency, they observed decreased expression levels and altered assembly patterns (i.e. accumulation of assembly intermediates) of the respiratory complexes I, II, and IV. In particular, they focused on complex I, for which they present extensive complexome profiling data of very high quality. Collectively the data show severe alterations in the assembly pathway of complex I, the largest of the respiratory complexes. They concluded that the assembly of individual respiratory complexes is interdependent (e.g. complex I assembly requires simultaneous assembly of complex III) and that complex III is the master regulator of the respiratory chain assembly.

Although the work is interesting and the figures are informative and well prepared, I do not think that the central conclusion is supported by the data for the following two reasons:

1. While the paper unequivocally documents inhibition of the complex I assembly pathway in the CYB-deficient cell line, it is not clear whether this inhibition is due to the general reduction of the expression levels of complex I subunits (or necessary assembly factors) or whether it is in fact the direct result of the lack of complex III.
2. Even if we are taking for granted that the assembly of individual complexes is interdependent, it does not mean that complex III is the "master regulator" because mutations in other complexes were not tested.

1st Revision - authors' response

17th October 2019

POINT BY POINT RESPONSE TO REFEREES:

Referee #1:

This work provides evidence for interplay between the redox complexes in terms of biogenesis.

I only have one point of criticism. In the beginning of the Introduction:

"..with the translocation of four protons from the mitochondrial matrix to the intermembrane space."

referring to complex III. This is erroneous and must be corrected.

We apologize for the lack of accurateness. We have now changed that sentence in the introduction to: "The mitochondrial respiratory chain (MRC) Complex III (cIII) or bc₁ complex is a trans-inner-membrane enzyme that couples the transfer of electrons from ubiquinol (reduced coenzyme Q or CoQ) to cytochrome c with proton translocation from the mitochondrial matrix to the intermembrane space, by means of the Q-cycle catalytic mechanism (Trumpower, 1990)."

Referee #2:

In this manuscript, M. Protosoni and co-workers study a cell line with a mutation in MT-CYB that results in deficiency of the mitochondrial respiratory chain complex III. Deficiencies in complex III have been previously reported to cause complex I deficiencies and the reported cell line shows a specifically strong reduction in the latter enzyme. It was previously assumed that this interdependency is indirect and that complex I decrease is caused by a destabilization of the fully assembled enzyme. In contrast, the current work shows that it is instead the failure to complete the assembly at a late stage. This late stage occurs by the addition of the NAD

dehydrogenase module (N-module), which according to the here-proposed model needs to occur in an association with complex III in a supercomplex. If true, this model would resolve long-standing conundrums relating to the independency of respiratory chain complex assembly and the role(s) of respiratory supercomplex formation. Overall, the manuscript clearly describes a set of high quality data obtained from the study of the D4-CYB cell line. It is, however, clear from the presented data that this mutation has very severe consequences for mitochondrial fitness and impacts on all respiratory chain enzymes.

We thank the Referee for the positive comments about our work and for the suggested experiments that have very much improved the manuscript.

In order to substantiate the proposed model, the authors should address experimentally two major points:

1. A key finding of this manuscript is that the assembly factor NDUFAF2 is highly upregulated in the D4-CYB cell line and binds to the late complex I assembly intermediate. Because NDUFAF2 occupies a position similar to the N-module, it is possible that the molecular explanation for the inability to complete complex I assembly are the increased levels of the assembly factor in the D4-CYB cell line. This could impair progression of assembly by blocking (parts of) the N-module binding sites. Such a hypothesis would be in line with data showing that complex I assembly can progress in absence of NDUFAF2 (PMID: 23702311). Hence, the authors should (1) deplete NDUFAF2 from the D4-CYB cell line by siRNA or similar approaches, and (2) overexpress NDUFAF2 in wild type cells, and test whether this either restores or impairs assembly, respectively.

To reply to this point, we have both overexpressed and knocked down NDUFAF2 expression in the two cell lines (WT and Δ 4-CYB). We have added these results in the new Figure EV7 in the Expanded View (Supplemental material) document. In agreement with Schlehe et al., knock-down expression of NDUFAF2 had no drastic effects on cI assembly or activity in the WT cells. In addition, depletion of NDUFAF2 did not promote cI maturation in the Δ 4-CYB cells (Figure EV7 D-F). On the other hand, overexpression of NDUFAF2 in the WT cells did not block cI maturation, showing the same amount of assembled active cI as the non-overexpressing cells (Figure EV7 A-C). Therefore, we can conclude that the observed block in assembly in the Δ 4-CYB mutant is not caused by the overexpression of NDUFAF2, but it could instead be a consequence of pre-cI stabilization, due to the impaired cI maturation. We have included a sentence in page 12 of the manuscript summarizing this additional information: "Given their sequence homology, NDUFAF2 probably occupies the binding site of the N-module subunit NDUFA12 in the cI structure, and its upregulation in Δ 4-CYB mitochondria could induce cI assembly stalling at the pre-cI stage by preventing the incorporation of the N-module. However, NDUFAF2 overexpression did not prompt the accumulation of pre-cI in the WT cells, and the amounts of mature active cI were the same as in the cells transfected with an empty vector (Figure EV7 A-C). In addition, downregulation of NDUFAF2 expression had no drastic effects on cI assembly or activity in the WT cells, as previously described (Schlehe et al., 2013), and did not promote cI maturation in the Δ 4-CYB cells (Figure EV7 D-F).".

2. In fungi assembly of complex I is not inhibited by mutations abolishing supercomplex formation due to absence of complex III and/or complex IV (PMID: 19111556 and references therein). A key feature of these fungi is that they can upregulate the expression of alternative oxidase (AOX), which directly oxidises reduced coenzyme Q, thereby alleviating the oxidative stress load of mitochondria. Consequently, it is possible that the molecular reason why complex III deficiency does not provoke loss of complex I activity is the presence of AOX in these fungi. Therefore, the authors should express AOX in the D4-CYB cell line to test whether reducing the accumulation of reduced coenzyme Q corrects complex I activity in a similar fashion. Expressing AOX would also allow correcting the respiratory deficiency of the D4-CYB cell line and comparing these cells better to the respiratory competent wild type cells.

Following the Referee's excellent suggestion, we expressed a tagged version of Emericella nidulans AOX (AOX_{HA}), shown previously to be functional in mouse cultured cells (PMID: 19020091 and 27052170), in both WT and Δ 4-CYB cybrids. CI maturation was improved in the Δ 4-CYB cybrids but only partially, as a significant amount of pre-cI remained stabilized by NDUFAF2 in the Δ 4-CYB AOX_{HA} cells (shown in the new Figure 7D). Also, cI enzymatic activity was increased significantly in the Δ 4-CYB AOX_{HA} cells, but it only reached 55% of the control values (new Figure 7F). In these cells the reduced to total CoQ ratios were most likely restored to physiological levels, because the Δ 4-CYB AOX_{HA} cells were able to grow in medium without uridine, contrary to the same cells transfected with an empty vector (new Figure 7B). Therefore, the new results suggest that there are two components in the dependency of cI biogenesis on cIII₂: functional and structural. The functional component, i.e. oxidation of the CoQ pool, can be partially overcome by AOX. However, in order to achieve efficient maturation of human cI, cIII₂ must be physically present. Also, the fact that inhibiting cIII₂ activity by treating the WT cells with antimycin A for a week did not significantly affect cI bound to the supercomplexes (new Figure 7I), indicates a strong structural component for cIII₂ binding during cI maturation.

We have added a whole new sub-section in Results describing these data (pages 15 to 16) and a paragraph in the Discussion section (page 20).

Minor points:

1. Complex II should be analysed also by BN-PAGE with western blotting. While the activity measurements do not indicate decreased levels, the complexome and steady state data show that also complex II is reduced. This is an important experiment because complex II, which is not part of the supercomplexes, could represent a good control for the specificity of the assembly defects.

We have now included additional images in Figure 1 (panel C) showing immunodetection of complex II and complex V of WB from BN-PAGE gels. In addition to the Δ 4-CYB clone that was used for all the analyses shown in the manuscript (#17.3E), we have added another clone with the same homoplasmic mutation (#17.3B) in order to rule out clone-specific differences in OXPHOS complexes levels.

We have also added the description of these results in Page 5 of the manuscript: “The profound reduction in cI amounts and activity of $\Delta 4$ -CYB was confirmed by in-gel activity assays (IGA) and by immunodetection, with a specific antibody against the cI subunit NDUFS1, following Blue-Native gel electrophoresis (BNGE) separation of the native MRC complexes in mitochondrial extracts from the $\Delta 4$ -CYB and WT cell lines (Figure 1B and C). The amounts of assembled complexes V and II were not drastically affected by the MT-CYB mutation in two different $\Delta 4$ -CYB clones: #17.3E (E) and #17.3B (B) (Figure 1C).”.

2. The authors should expand their mechanistic discussions in regards to the recent structural work. Can they speculate further how the binding of complex III to complex I facilitates the finalization of assembly?

As discussed in the Major Point 2, in view of the new data, we now propose the idea of a double functional and structural role for $cIII_2$ to promote cI assembly. As requested, we have speculated how these actions could be exerted during SC biogenesis. The functional component would presumably act through reduced CoQ and we have speculated that this could reduce the Fe-S clusters, which must be present already in the pre-cI structure, creating a highly reduced environment and changing the polarity. This could possibly impede the incorporation of the N-module through electrostatic repulsion. Given that there is no direct contact of $cIII_2$ with the N-module of cI in the respirasome architecture, the structural component would imply some kind of conformational change of the peripheral arm when $cIII_2$ binds, allowing the exchange of NDUFAF2 for the N-module. The existence of a structural cross-talk between cI and $cIII_2$ was recently described (PMID: 31492636). Also, we have performed a rough alignment between the ovine SC structure and the bovine free cI, and they clearly differ in the relative position of subunits in the Q-module, immediately adjacent to the N-module. We have now introduced these ideas in the Discussion (page 20).

Referee #3:

In this manuscript, Protasoni et al studied the assembly of respiratory complexes in a mutant cell line deficient in MtCYB, a subunit of complex III encoded by mitochondrial DNA. As a result of CYB deficiency, they observed decreased expression levels and altered assembly patterns (i.e. accumulation of assembly intermediates) of the respiratory complexes I, II, and IV. In particular, they focused on complex I, for which they present extensive complexome profiling data of very high quality. Collectively the data show severe alterations in the assembly pathway of complex I, the largest of the respiratory complexes. They concluded that the assembly of individual respiratory complexes is interdependent (e.g. complex I assembly requires simultaneous assembly of complex III) and that complex III is the master regulator of the respiratory chain assembly.

Although the work is interesting and the figures are informative and well prepared, I do not think that the central conclusion is supported by the data for the following two reasons:

We thank the Referee for the positive view about this work and the chance to discuss the findings further.

1. While the paper unequivocally documents inhibition of the complex I assembly pathway in the CYB-deficient cell line, it is not clear whether this inhibition is due to the general reduction of the expression levels of complex I subunits (or necessary assembly factors) or whether it is in fact the direct result of the lack of complex III.

To answer the Referee's point we have measured mRNA levels of several complex I subunits belonging to different structural modules of the complex, and both encoded in the mtDNA (MT-ND1) and in the nuclear genome (NDUFB8 and NDUFB11). As it can be seen in the graph below, there was not a reduction at the transcript level, ruling out a lower expression of these subunits, at least at the transcriptional level.

Figure R1: *Quantitative RT-PCR analysis of the expression at the mRNA level of three structural complex I subunits. Total RNA was extracted from WT and Δ4-CYB cells and was retrotranscribed. The expression levels were measured using the specific TaqMan Gene Expression Assays (Applied Biosystems, Thermo Fisher Scientific) for each transcript. The expression levels of each subunit were normalized to that of the housekeeping gene GAPDH.*

In addition, the quantitative proteomics analysis shown in Figure 2A indicated no change in the protein levels of several complex I assembly factors. The figure below shows a graph generated with the same data as in Figure 2A, showing only the detected complex I assembly factors. Only NDUFAF2 showed a significant increase (over the threshold of log₂ ratios of 2 or under the log₂ ratios of -2).

Figure R2: Quantitative proteomics analysis showing the relative levels of the detected known complex I assembly factors. Scatter plot generated from the peptide content analyzed by mass spectrometry in each of the 64 slices excised from BNGE and after quantifying the heavy to light (H/L) and (L/H) ratios in both reciprocal labeling experiments performed with mitochondria isolated from WT and $\Delta 4$ -CYB cells.

To make this point clearer in the manuscript, we have added a sentence in page 6 that reads: “The amounts of nine known cI assembly factors (ACAD9, ECSIT, FOXRED1, NDUFAF1, NDUFAF2, NDUFAF3, NDUFAF4, NDUFAF6 and TMEM126B), did not differ significantly between the $\Delta 4$ -CYB vs. WT cells (Figure 2A).”

Additional evidence ruling out lower expression of the complex I structural subunits in the $\Delta 4$ -CYB cells is shown in Figures 2B and 6A. These translation assays revealed that the synthesis of the seven mtDNA-encoded cI subunits is not reduced in the $\Delta 4$ -CYB mutant. To further clarify this issue, we have added a sentence in page 6 of the Results: “This analysis indicated that only MT-CYB was not being translated in the $\Delta 4$ -CYB cybrids and that there was no clear reduction in the synthesis of any of the seven ND (cI) subunits (Figure 2B).”.

2. Even if we are taking for granted that the assembly of individual complexes is interdependent, it does not mean that complex III is the "master regulator" because mutations in other complexes were not tested.

According to the current literature, structural alterations of the mitochondrial respiratory chain (MRC) complexes have major diagnostic implications in human disorders, since mutations specifically targeting one given complex frequently alter the activities of the remainder MRC complexes. Mutations severely affecting complex III structure, like the one described in our manuscript, consistently cause a parallel complex I (and in some cases also complex IV) enzymatic defect in patients and disease models (to name a few PMIDs: 15208329, 25914718, 28275242, 23910460, 28804536, 25008109, 16740593, 22106410, 15053874), and in this work we explain the molecular mechanism involved. It is true that in this manuscript we have not addressed the question using other mutated cell lines. However, mounting data available in the literature and our own unpublished work indicate that severe complex IV structural defects necessarily promote different pathophysiological responses, as they commonly present in patients as isolated cIV deficiency, being the pleiotropic effects on complex I or complex III extremely rare (PMIDs: 30030362, 26846578). Few exceptions involve homoplasmic nonsense mutations leading to the structural and functional loss of the mitochondrial-encoded subunit MT-CO1, which have been associated with combined enzyme deficiencies of cI and cIV exclusively in cybrid cell lines (PMIDs: 16740593, 22106410, 17452320). Using high-throughput proteomics and comprehensive biochemical analyses in these complex IV mutants, we (see Figure R3 below) and others have demonstrated that the loss of MT-CO1, and complete holo-cIV, still permits the full assembly of complexes I and III₂ and their binding to form supercomplex I+III₂ (PMIDs: 16740593, 27545886, 22252130); however, the absence of cIV results in a lower stability of this SC and complex I dissociation, both structures getting degraded through the action of the m-AAA protease AFG3L2 (PMID: 22252130). Therefore, this implies that complete and efficient complex I assembly would mainly rely on its association with complex III within the SC I+III₂, which would require the binding of fully-assembled cIV to maintain its stability.

Figure R3: Figure for Referees not shown.

In addition, mutations that completely impair complex I assembly barely affect the assembly of complex III₂ or complex IV, which still associate with each other in the III₂+IV SC (see for example PMID: 27626371).

*In summary, considering that differently to the situation described in this paper, lack of cIV permits cI maturation and that without cIII₂ there is no formation of any SC species, we determined that cIII₂ has an indispensable role in the regulation of MRC assembly, and in particular in that of cI. However, we agree that calling cIII₂ the "master regulator" might be a strong statement at this stage and with the data provided for publication. Consequently, we have modified the Abstract substituting "... complex III is the **master regulator** of MRC maturation..." for "... complex III is **central** for MRC maturation...". Also, we have softened the statement in the conclusion paragraph (page 22), changing the sentence where it said that the SCs*

are complex III-regulated platforms for the assembly of the MRC. Now it reads that complex III is central in these structures and in the process of MRC assembly.

2nd Editorial Decision

29th October 2019

Thank you for submitting a revised version of your manuscript. It has now been seen by the original referees whose comments are shown below.

As you will see, they find that all criticisms have been sufficiently addressed and recommend the manuscript for publication. However, there are a few editorial issues concerning text and figures that I need you to address before we can officially accept the manuscript.

REFEREE REPORTS

Referee #1:

The only criticism I had has been adequately dealt with.

Referee #2:

In the revised version of the manuscript, Protasoni et al have addressed all the points raised by this reviewer. The additional experiments have strengthened the manuscript and clarified critical points. In summary, this is a very interesting, concise manuscript reporting a very important finding for mitochondrial biogenesis.

Referee #3:

I am recommending to accept the manuscript

Corresponding Author Name: Erika Fernandez-Vizarra

Journal Submitted to: The EMBO Journal

Manuscript Number: EMBOJ-2019-102817